# A self-stabilized and water-responsive deliverable coenzyme-based polymer binary elastomer adhesive patch for treating oral ulcer

Chunyan Cui[1,4], Li Mei[2,4], Danyang Wang[2], Pengfei Jia[2], Qihui Zhou [3] ✉ & Wenguang Liu [1] ✉

Oral ulcer can be treated with diverse biomaterials loading drugs or cytokines. However, most patients do not benefit from these materials because of poor adhesion, short-time retention in oral cavity and low drug therapeutic efficacy. Here we report a self-stabilized and water-responsive deliverable coenzyme salt polymer poly(sodium α-lipoate) (PolyLA-Na)/coenzyme polymer poly(α-lipoic acid) (PolyLA) binary synergistic elastomer adhesive patch, where hydrogen bonding cross-links between PolyLA and PolyLA-Na prevents PolyLA depolymerization and slow down the dissociation of PolyLA-Na, thus allowing water-responsive sustainable delivery of bioactive LA-based small molecules and durable adhesion to oral mucosal wound due to the adhesive action of PolyLA. In the model of mice and mini-pig oral ulcer, the adhesive patch accelerates the healing of the ulcer by regulating the damaged tissue inflammatory environment, maintaining the stability of oral microbiota, and promoting faster re-epithelialization and angiogenesis. This binary synergistic patch provided a therapeutic strategy to treat oral ulcer.

Oral mucosal injury (oral ulcer), a common and high-incidence oral disease, is manifested as persistent loss or destruction of the oral epithelium. More than 25% of people worldwide have experienced or are experiencing oral ulcers[1–3]. When oral ulcers are not treated promptly or adequately, the epithelial tissue in the ulcer falls off and form depressions, or even necrosis occurs, making it difficult for patients to chew, swallow, speak, or even digest[3,4]. In addition, oral wounds are susceptible to bacterial infection due to the loss of the inherent protective function of the oral mucosa, which can further slow down the healing efficiency of oral ulcers[5,6]. Therefore, accelerating the healing of oral ulcer is critical to the maintenance of oral health. Similar to cutaneous wound healing, oral ulcer healing usually consists of three sequential but overlapping phases, namely the

inflammatory phase, cell proliferation phase, and tissue remodeling phase[7–9]. Although some commercial oral ulcer patches (chitosan, flavonoids, etc.), powders (vitamins, etc.) and ointments (recombinant human epidermal growth factor gel, etc.) have been developed to accelerate oral ulcer healing, they suffer from low treatment efficiency due to their short retention time on the surface of mucosa in the highly humid and dynamic environment of the oral cavity (<2 h)[10,11]. To address this limitation, wet oral mucosa adhesive patches with the three-dimensional (3D) network are designed by used 3D printing technology, electrospinning technology, photocontrolled gelation technology and drying film technology, and widely used as a platform to allow for the controlled release of therapeutic drugs, cytokines and protein to accelerate the regeneration and remodeling of the oral

[1]School of Materials Science and Engineering, Tianjin Key Laboratory of Composite and Functional Materials, Tianjin University, Tianjin 300350, China. [2]Department of Stomatology, Qingdao University, Qingdao 266021, China. [3]School of Rehabilitation Sciences and Engineering, University of Health and Rehabilitation Sciences, Qingdao 266071, China. [4]These authors contributed equally: Chunyan Cui, Li Mei. ✉e-mail: qihuizhou@qdu.edu.cn; wgliu@tju.edu.cn

mucosa[12–18]. These mucodhesive patches successfully extended the residence time of the patch in the oral cavity to 3–10 h based on the improved structural stability and multiple interactions with mucosal tissue, which effectively improved the release efficiency of the therapeutic drugs, cytokines and proteins at the site of mucosal injury. In particular, several studies have shown good therapeutic effects on human volunteers and clinical trials have been carried out, which are very encouraging[19,20], However, most of these oral mucoadhesive patches often require complex preparation techniques. In addition, they lack intrinsic therapeutic efficacy, and exogenous drugs, cytokines or proteins load will increase the cost of the patch or may cause potential drug toxicity. Therefore, this encourages us to use simple preparation methods to explore a self-therapeutic and low-cost bioactive mucoadhesive patch to accelerate oral mucosal injury.

For the repair of oral mucosal injury, it is highly desired for the oral mucoadhesive patch to self-regulate the wound microenvironment (inflammation, bacterial infection, and host cell behavior) and protect the injured interface in the saliva environment for a long time (>12 h, this is the optimal repair time for treating oral mucosal disorders[21]) to meet the demand of the oral mucosa regeneration. In recent years, an array of bioactive materials, which may effectively accelerate tissue regeneration by anti-inflammation, sterilization, and promoting the secretion of growth factors without loading drugs, cells or cytokines, have been developed and applied to epidermal wound healing[22–25]. However, those bioactive materials that have been reported lack durable adhesion ability and stability in wet environment. In addition, the complex structural design and functional modifications and the external interventions may hinder their clinical translation[21].

α-Lipoic acid (LA) is a coenzyme found in mitochondria, and participates in mitochondrial activity and synergistic energy metabolism[26,27]. Moreover, LA shows an excellent antioxidant ability by scavenging ROS, coupling metal ions and regenerating endogenous antioxidants[27]. Previous studies have shown that LA also has antibacterial activity by changing the permeability of the cell membrane[28,29]. In addition, based on its ring-opening self-polymerization (ROP) mechanism initiated by dynamic disulfide bond exchange and abundant adhesive carboxyl groups, LA has been employed to prepare supramolecular polymer adhesives[30–33]. However, the as-polymerized poly(lipoic acid) (PolyLA) is metastable due to the inverse ring closing depolymerization initiated by terminal reactive radicals[30]. Although it has been reported that PolyLA-based polymers can be stabilized by introducing multiple double-bond monomers, hydrogen bond interactions and ionic complexation, the obtained PolyLA-based polymers are difficult to be purified, and the unreacted foreign molecules that cannot be removed pose a potential risk to organisms[30,31,34,35]. It is necessary to point out that the stabilized PolyLA and the natural hydrophobicity of LA restrict the release of bioactive LA monomer when used in vivo, sacrificing its biofunctions. Therefore, the effect of released LA on tissue repair has not been well harnessed. In contrast, sodium α-lipoate (LA-Na) is amphiphilic and can not only spontaneously open rings and self-assemble into a highly ordered PolyLA-Na network, but also exhibit a water-sensitive dissociated behavior[28]. When PolyLA-Na comes into contact with water, it can be quickly dissociated into small LA salts[36]. However, the excessively rapid dissociation rate (less than 20 min), non-adhesive and rigid nature of PolyLA-Na greatly limit its direct application in the biomedical field.

In this study, aiming at highly efficient therapy of oral ulcer, we will develop a self-stabilized and water-responsive deliverable PolyLA-based elastomer adhesive patch. The core idea is to create a PolyLA-Na/PolyLA binary synergistic system, where the hydrogen bonding cross-links between PolyLA and PolyLA-Na prevent PolyLA depolymerization, and thus prolongs the release time of bioactive LA-Na molecules, which is crucial for long-term and effective scavenging of ROS from the wound site. While the amorphous PolyLA serves to reduce the crystallinity of PolyLA-Na, thus increasing the elasticity and extensibility of the PolyLA-

Na film. Meanwhile, this PolyLA-Na/PolyLA binary elastic and extensible patch exhibits robust wet tissue adhesion due to the adhesive action of PolyLA, whose carboxyl groups can achieve rapid and firm adhesion to tissue by forming multiple hydrogen bonds and electrostatic interactions with the amino groups on the tissue surface[30]. And dried hydrophilic PolyLA-Na network can remove the hydration layer by water adsorption. It is anticipated that although the water-responsive PolyLA-Na is gradually dissociated in the wet environment, the adhesive PolyLA hydrophobic network is still maintained due to the multiple hydrogen bonding interactions between carboxyl groups in its polymer network and its hydrophobic nature. As a result, the PolyLA-Na/PolyLA adhesive elastomer patch can keep a long-term wet adhesion in vivo. This binary synergistic adhesive is used as an oral ulcer repair patch, and its long-lasting wet-resistant adhesion to tissue can effectively prevent wound infection caused by food residue and microorganism, while the continuous release of the therapeutic small molecule can significantly accelerate oral mucosal tissue regeneration and remodeling by eliminating harmful bacterial infection and excess reactive oxygen species (ROS). Taken together, this PolyLA-Na/PolyLA binary synergistic elastomer adhesive patch holds a great clinical application prospect in accelerating the repair of oral ulcer (Fig. 1).

## Results

### Preparation and characterization of PolyLA-Na/PolyLA binary synergistic adhesive patches

Previous reports have shown that intermolecular disulfide exchange occurs when the LA molecules are close enough to each other in ethanol at a high concentration, resulting in the formation of PolyLA with alternating disulfide bonds in the backbone[34]. While LA-Na can also undergo spontaneous ring-opening polymerization at a higher concentration in an aqueous solution[36]. Based on these mechanisms, we designed and fabricated a self-stabilized water-responsive deliverable PolyLA-Na/PolyLA binary synergistic adhesive patch by one-pot mixing of the PolyLA ethanol solution and PolyLA-Na aqueous solution and allowing the solvent to evaporate sufficiently at 37 °C (Supplementary Fig. 1). Firstly, we investigated the spontaneous ring-opening and polymerization ability of LA and LA-Na in ethanol and water, respectively. UV-vis spectra show that the absorption peak intensity of the disulfide-containing five-membered ring in LA monomer at 330 nm decreases with the increase of the concentration of LA in ethanol, and the PolyLA peak appears at 260 nm (Supplementary Fig. 2a), indicating that LA indeed undergoes ring-opening polymerization spontaneously at higher concentrations in ethanol[34,35]. The significantly increased viscosity of LA in ethanol also suggests that LA has polymerized (Supplementary Fig. 2b). [1]H nuclear magnetic resonance (NMR) spectroscopy confirms the ROP of LA monomer in ethanol again (Supplementary Fig. 3) due to the emergence of a new peak of disulfide bonds at 2.8 ppm[30,31,34]. However, although PolyLA is stable in ethanol, after ethanol volatilization, PolyLA becomes unstable again and becomes an opaque and non-adhesive film (Fig. 1a and Fig. 2a). Likely, spontaneous ring-opening polymerization of LA-Na in aqueous solution also occurs at high concentrations (Fig. 1b, Supplementary Fig. 4, 5). However, unlike PolyLA in an ethanol solution, the PolyLA-Na further self-assembles into a highly ordered supramolecular network with the evaporation of the solvent[36], and the film formed is translucent (Fig. 2a) due to its semi-crystalline nature, which will be confirmed by XRD pattern in the following experiment. However, due to its high sensitivity to water, the PolyLA-Na film can be rapidly dissociated in a water environment (Supplementary Fig. 6). Then, we investigated the physicochemical properties of PolyLA-Na/PolyLA binary system, and found that this binary system circumvented the metastability of PolyLA and the fast degradation as well as the rigidity of PolyLA-Na, and new adhesive function, high extensibility and sustainable release behavior of bioactive LA-Na emerged. In this work, it is noted that the concentration of LA-Na was set at 1.5 mmol/mL, and the concentration of LA and the ratio of LA/LA-Na

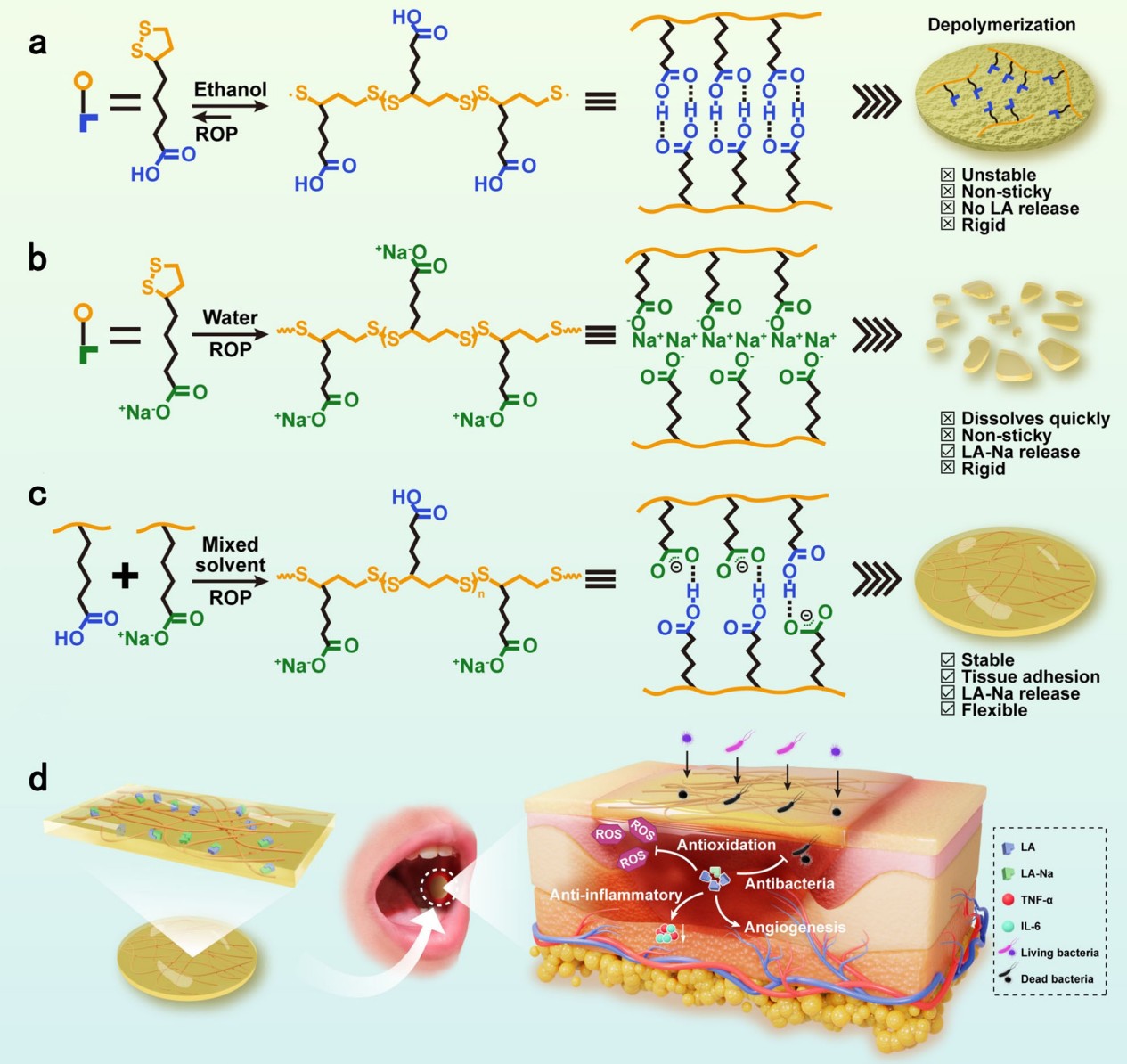

**Fig. 1 | Schematic diagram of preparation and application of polyLA-based adhesive patch. a** Preparation process and structural characteristics of PolyLA; **b** Preparation process and structural characteristics of PolyLA-Na patch; **c** Preparation process and structural characteristics of PolyLA-Na/PolyLA binary synergistic adhesive patch; **d** Illustration of the proposed mechanism of PolyLA-Na/PolyLA patch for promoting oral ulcer healing. LA α-lipoic acid, LA-Na sodium α-lipoate, TNF-α tumor necrosis factor-α, IL-6 interleukin-6.

were adjusted to obtain a series of elastic patches. The resultant patches are named PolyLA-Na/PolyLA-X-Y, where X denoted the concentration of LA in ethanol, and Y denotes the volume ratio of LA to LA-Na; for example, PolyLA-Na/PolyLA-1.5-0.5 means that the concentration of LA in ethanol is 1.5 mmol/mL, the concentration of LA-Na in water is 1.5 mmol/mL, and the volume ratio of LA /LA-Na is 0.5:1.

The characteristic Raman peak of disulfide bond in the monomer structure of LA and LA-Na at 511 cm$^{-1}$ is split into two peaks at 509 and 524 cm$^{-1}$, suggesting that the sulfur-containing five-membered rings in LA and LA-Na monomers have undergone ring-opening polymerization in the PolyLA-Na/PolyLA binary system, and both the formed PolyLA and PolyLA-Na are stable[30,34,37] (Fig. 2b). The interaction mechanism of PolyLA and PolyLA-Na in the binary system was investigated by FT-IR spectra (Fig. 2c). The -OH peak at 1249 cm$^{-1}$ on the carboxyl group disappears in the PolyLA-Na film, and the -C=O peak shifts from 1694 to 1561 cm$^{-1}$, indicating that all carboxyl groups in PolyLA-Na are

deprotonated[36]. While the appearance of the -OH peak at 1249 cm$^{-1}$ also suggests the presence of carboxyl groups in this binary system. Compared to the pristine PolyLA-Na film (Supplementary Fig. 7), a slight red shift from 1561 to 1551 cm$^{-1}$ occurs to -C=O band on the carboxylate in the PolyLA-Na/PolyLA film, which is originated from the hydrogen bonding interaction between the carboxyl group and the carboxylate[36]. The hydrogen bonding interaction between PolyLA and PolyLA-Na can also be verified by $^{1}$H NMR spectra (Fig. 2d), where the -COOH in PolyLA-Na/PolyLA-1.5-1 shifts to a lower chemical shift compared to PolyLA, suggesting the existence of -COOH···O=C- hydrogen bond[34]. Molecular dynamics simulation was also performed to evaluate the total potential energy of different systems (Fig. 2e–g). Figure 2h exhibits that the total potential energy of PolyLA (−15043.78 kcal/mol) is greater than that of PolyLA-Na (−26593.67 kcal/mol) and PolyLA-Na/PolyLA adhesive (−44418.26 kcal/mol), implying the formation of thermodynamically stable -COOH···O=C- hydrogen bond, which lowers the potential energy

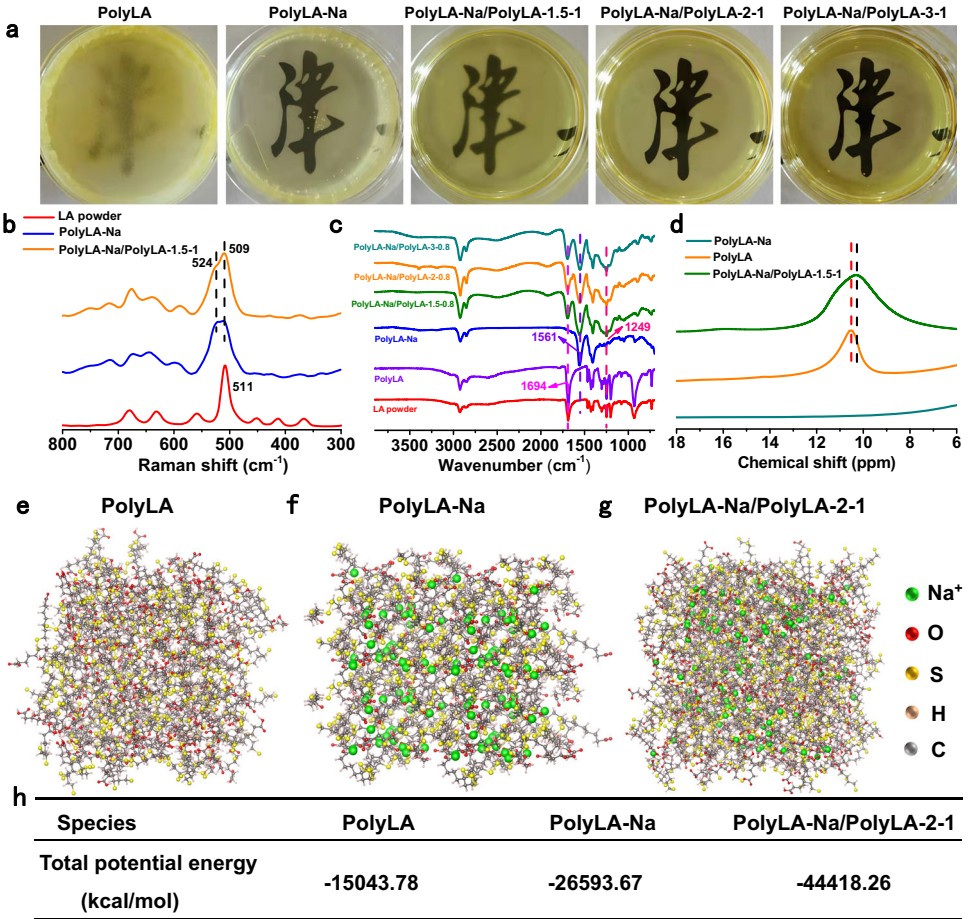

**Fig. 2 | Characterizations and molecular dynamics simulation of PolyLA-Na/ PolyLA binary synergistic patches. a** Pictures of dry PolyLA, dry PolyLA-Na film and PolyLA-Na/PolyLA patches with different compositions; **b** Raman spectra of the LA powder, dry PolyLA-Na film, and PolyLA-Na/PolyLA-1.5-1 patch; **c** FT-IR spectra of the LA powder, dry PolyLA, dry PolyLA-Na film and PolyLA-Na/PolyLA patches with different compositions; **d** $^1$H NMR spectra of dry PolyLA, dry PolyLA-Na, and PolyLA-Na/PolyLA-1.5-1 patch; **e–g** Molecular dynamics simulation of potential energy of PolyLA, PolyLA-Na and PolyLA-Na/PolyLA-2-1 systems; **h** Total potential energy value of PolyLA, PolyLA-Na and PolyLA-Na/PolyLA-2-1 systems.

of PolyLA and PolyLA-Na to form a stable PolyLA-Na/PolyLA binary system[34].

X-ray diffraction (XRD) was employed to illustrate structural changes in different polymers. Figure 3a shows that the sharp crystalline peaks of LA monomer appear in dry PolyLA (Fig. 3b) after ROP in ethanol, further indicating that PolyLA is metastable in ethanol[30,34]. The sharp and high-intensity diffraction peaks in the small angle regime (less than 20°) of dry PolyLA-Na film (Fig. 3c) suggest a typical, highly ordered structure at the nanoscale[29]. However, for the PolyLA-Na/PolyLA-based film, the crystallization peaks of pristine PolyLA disappears completely (Fig. 3d and Supplementary Fig. 8), meaning that the mixing of PolyLA and PolyLA-Na can stabilize PolyLA and inhibit the depolymerization of PolyLA; while the intensity of sharp diffraction peak of PolyLA-Na gradually decreases and eventually disappears with the increase of PolyLA content, indicating that amorphous phase is dominant in PolyLA-Na/PolyLA binary system. Comparing the XRD results of the PolyLA-Na/PolyLA-2-0.5 and the pristine PolyLA-Na film reveals that the diffraction angle of PolyLA-Na decreases after mixing with PolyLA, indicating that the combining PolyLA increases the interlayer distances of PolyLA-Na. (Fig. 3e)[36]. Polarized optical microscopy was also employed to detect the crystallization behavior of the binary system with different compositions. As shown in Supplementary Fig. 9, pristine PolyLA-Na film exhibits a bright cyan pattern, suggesting the formation of

structurally ordered assemblies[36], while the bright cyan gradually decreases with an increment of PolyLA content and transforms into a uniform, isotropic pattern, indicating dominant amorphous phase in the binary system. Similar results can also be illustrated by light microscopy (Supplementary Fig. 10), where a notable decrease in optical transparency is observed upon stretching PolyLA-Na/PolyLA-1.5-0.5 and PolyLA-Na/PolyLA-1.5-1, which is due to the tension-induced ordered arrangement of the polymer chains. With the increase of PolyLA content, the transparency of the stretched adhesive remains almost unchanged, indicating the formation of an amorphous PolyLA-Na/PolyLA binary system[36]. We calculated the crystallinity of the patch with different composition by using the XRD results. As shown in Supplementary Fig. 11, for pure PolyLA-Na, it is a highly crystalline structure with a crystallinity of 91%. After introducing PolyLA into PolyLA-Na system, the crystallinity of the patch decreased obviously, and the crystallinity of PolyLA-Na/PolyLA-1.5-0.5 was 13.5%. Further increasing the content of PolyLA, the crystallinity of the patch further decreased, and the crystallinity of PolyLA-Na/PolyLA-2-1 and PolyLA-Na/PolyLA-3-1 was only 1.4% and 0.7%, which proves once again that the PolyLA-Na/PolyLA-based patch gradually changed from crystalline dominant phase to amorphous dominant phase with the increase of PolyLA content.

Dry PolyLA-Na films are brittle and rigid due to the existence of crystalline structure (Supplementary Fig. 12a). It is noted that

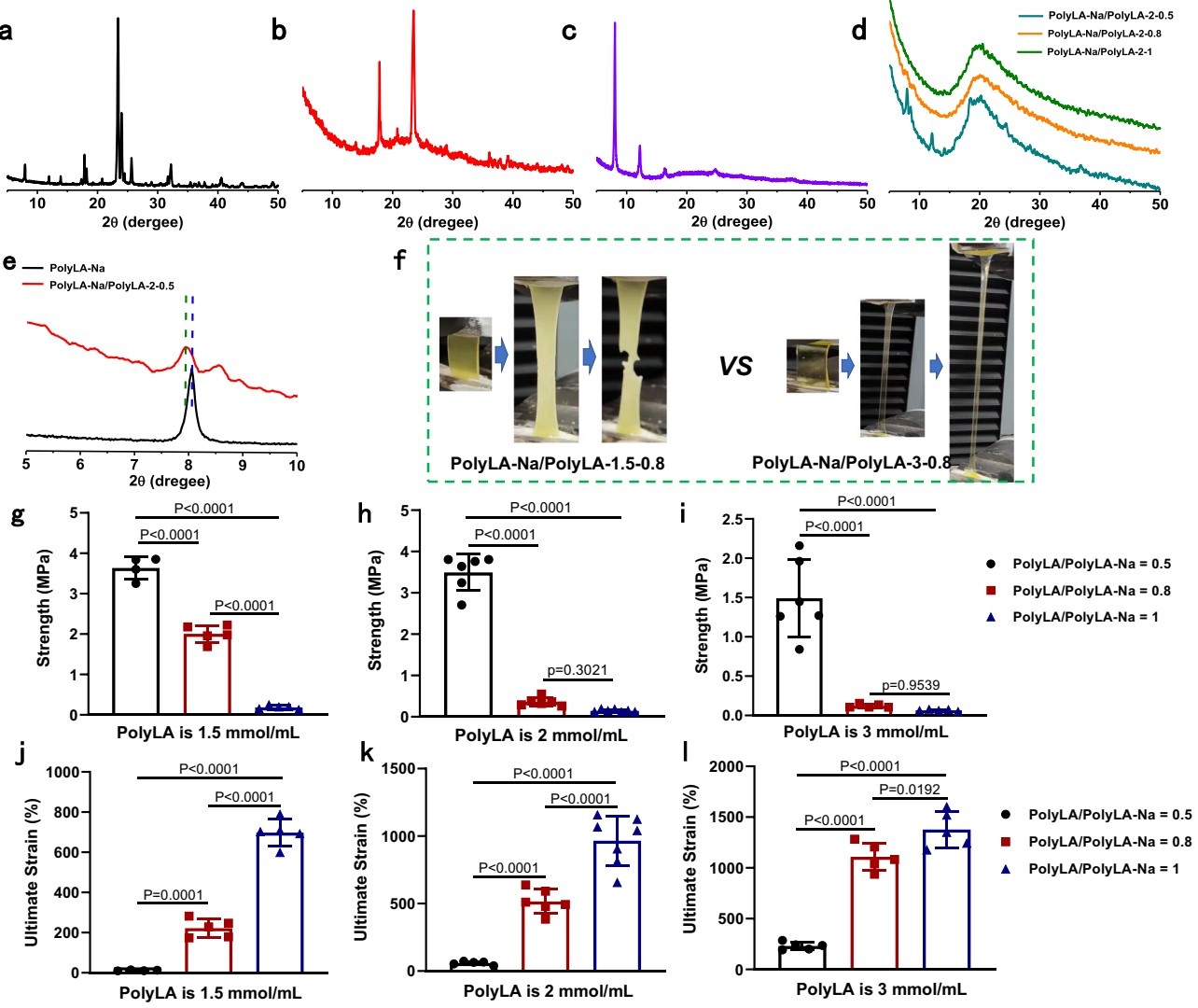

**Fig. 3 | Crystallization behavior and mechanical properties of PolyLA-Na/ PolyLA binary synergistic patches. a–e** XRD patterns of the LA powder, dry PolyLA, dry PolyLA-Na film, and PolyLA-Na/PolyLA patches with different compositions; **f** Pictures showing the stretching process of PolyLA-Na/PolyLA-1.5-0.8 and PolyLA-Na/PolyLA-3-0.8; **g–i** Tensile strength of PolyLA-Na/PolyLA patches with different compositions; **j–l** Ultimate strain of PolyLA-Na/PolyLA patches with different compositions **g, j:** $n = 4$ for PolyLA/PolyLA-Na = 0.5, $n = 5$ for PolyLA/ PolyLA-Na = 0.8 and 1; **h, k** $n = 6$ for PolyLA/PolyLA-Na = 0.5 and 0.8, $n = 7$ for PolyLA/PolyLA-Na = 1; **i, l** $n = 6$ for PolyLA/PolyLA-Na = 0.5, $n = 5$ for PolyLA/PolyLA- Na = 0.8 and 1. (All presented data are mean values ± SD. Statistics was calculated by one-way ANOVA followed by Tukey's post-test).

the films are too brittle to be tested in tensile loading. Comparatively, the tensile strength and modulus of the PolyLA-Na/PolyLA-based patch decrease significantly (tensile strength range is $3.63 \pm 0.21$–$0.074 \pm 0.01$ MPa, Young's modulus range is $3.05 \pm 0.6$–$0.0023 \pm 0.0008$ MPa) (Fig. 3g–i and Supplementary Fig. 13), while the elongation increases greatly with increasing polyLA content (elongation range is $14.6 \pm 2.3$–$1377 \pm 250\%$) (Fig. 3f, j–l). As shown in Supplementary Fig. 12b, PolyLA-Na/PolyLA-2-1 can be folded into a complex shape without breaking, indicating high flexibility. This result is attributed to the formation of dynamic hydrogen bonds between the PolyLA and the PolyLA-Na network, aiding in energy dissipation. From the tensile stress-strain curves of PolyLA-Na/PolyLA-based patches with different compositions (Supplementary Fig. 14), we can see that with increasing the content of PolyLA, the binary system transforms from a typical crystalline polymer to an amorphous polymer. Macroscopically, the binary system patch switches from translucent to highly transparent, implying the transition from crystalline to amorphous phase (Fig. 2a).

## In vitro wet tissue adhesion of PolyLA-Na/PolyLA patches

The oral mucosa is a moist and dynamic tissue, so developing a patch that can form stable and firm adhesion with the oral mucosa is crucial for the treatment of oral ulcers[21]. It is well known that an efficient drainage mechanism is a key to achieving adhesion to wet tissues[38,39]. As described above, the PolyLA-Na is highly water sensitive and its ordered hierarchical structure facilitates the transport of moisture[36], so when the PolyLA-Na/PolyLA adhesive patch is in contact with the wet tissue, the PolyLA-Na network can immediately absorb the liquid on the tissue surface and remove hydration layer. The rapid water absorption of the PolyLA-Na/PolyLA adhesive patch was confirmed by the water contact angle test. Taking PolyLA-1.5-0.5 as an example, the water contact angles on its surface gradually decrease with time, showing its rapid water adsorption capacity (Fig. 4a). The water contact angles of the PolyLA-Na/ PolyLA-based adhesive patches with different compositions were recorded after being in contact with water for 20 s (Fig. 4b and Supplementary Fig. 15). With the increase of PolyLA content, the water contact angles on the patch gradually increase, indicating enhanced

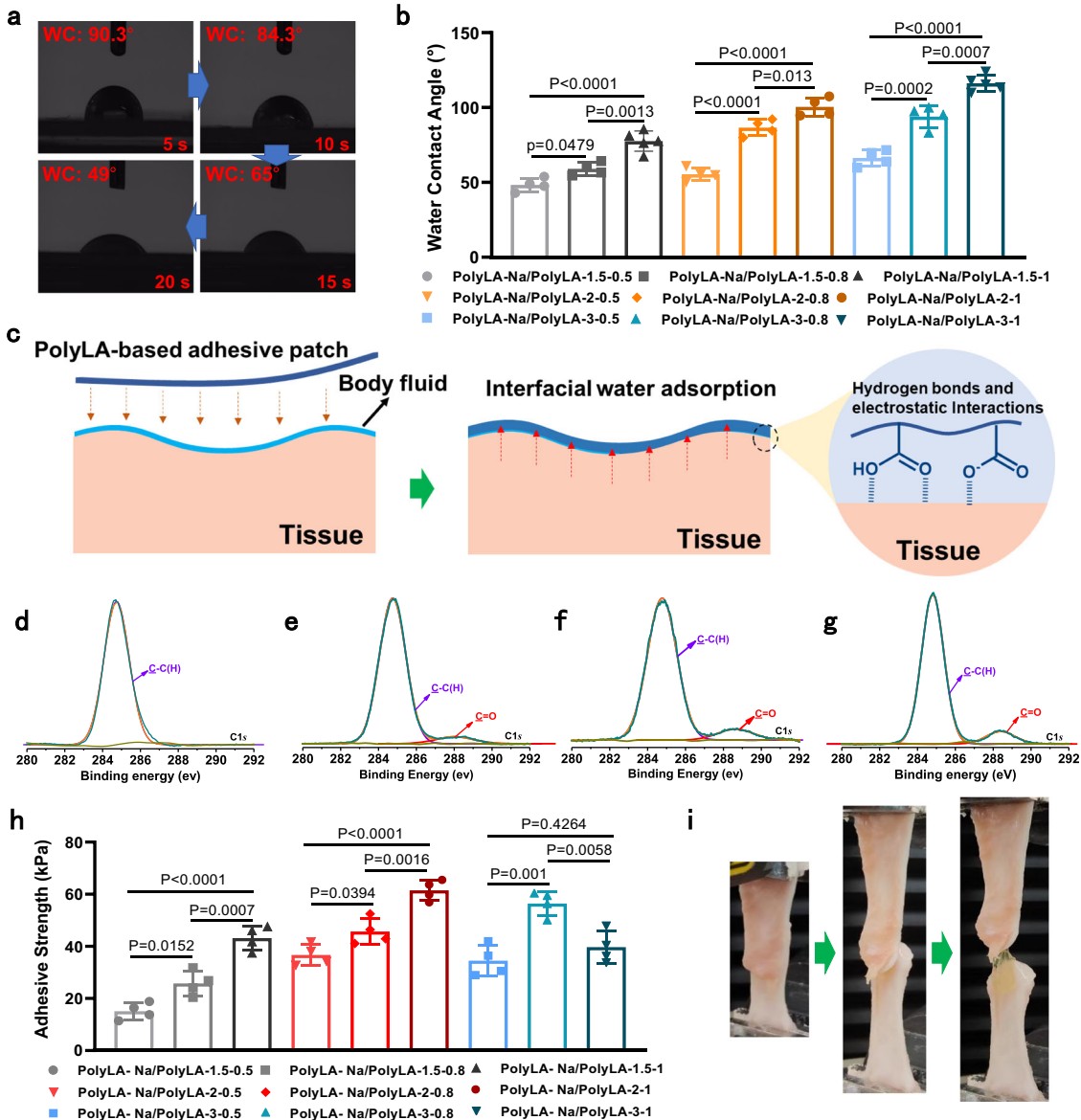

**Fig. 4 | Wet adhesion behavior of PolyLA-Na/PolyLA binary synergistic adhesive patches. a** Pictures showing the variation in the water contact angle of PolyLA-Na/PolyLA-1.5-0.5 patch at different times (WC: water contact angle); **b** The value of water contact angle of the PolyLA-Na/PolyLA adhesive patches with different compositions (*n* = 4); **c** Proposed adhesion mechanism of PolyLA-Na/PolyLA adhesive patch; **d–g** X-ray photoelectron spectra of dry PolyLA-Na film and PolyLA-Na/PolyLA adhesive patches with different compositions (**d** PolyLA-Na; **e** PolyLA- Na/PolyLA-1.5-1; **f** PolyLA-Na/PolyLA-2-1; **g** PolyLA-Na/PolyLA-3-1); **h** Adhesion strength of PolyLA-Na/PolyLA adhesive patches with different compositions to oral mucosa tissue after soaking in artificial saliva (*n* = 4); **i** Photographs showing the lap-shear process of the PolyLA-Na/PolyLA-2-1 patch adhered to oral mucosa tissue after soaking in artificial saliva. (All presented data are mean values ± SD from the mean from n = 4 independent measurements on independent samples. Statistics were calculated by one-way ANOVA followed by Tukey's post-test.)

hydrophobicity of the patch. Carboxyl groups can tightly bind with tissues by forming multiple hydrogen bonds and electrostatic interactions with tissue surface (Fig. 4c)[40,41]. X-ray photoelectron spectroscopy results show that after mixing PolyLA, carboxyl appears on the surface of the adhesive patch, and the number of carboxyl groups increases with the increment of PolyLA content (Fig. 4d–g). To simulate the oral environment, we performed adhesion strength tests on pig oral musoca tissue soaked in artificial saliva. The results show that PolyLA-Na/PolyLA adhesive patch can adhere to the artificial saliva-treated oral musoca tissue instantly, and the highest instant adhesion strength can reach 60 kPa (Fig. 4h and Supplementary Fig. 16), and the patch still maintains firm adhesion even if its bulk is stretched during the test (Fig. 4i). For PolyLA-Na/PolyLA-1.5-Y and PolyLA-Na/PolyLA-2-Y patches, their tissue adhesion strengths are enhanced with the PolyLA content, because the

augment of carboxyl group content on the surface leads to enhanced hydrogen bonding and electrostatic interactions between the patch and the tissue. However, for PolyLA-Na/PolyLA-3-Y, the adhesion strength shows a trend of increase first and then decrease with the increment of PolyLA content. The reason is that although the content of the carboxyl group on the surface of the patch increases, the excessive PolyLA leads to the decreased bulk strength and drainage performance of the patch, thus affecting its tissue adhesion. Currently commercially available oral ulcer treatment patches or drugs can only stay on the lesions for a very short-time in the salivary environment. While durable adhesion of the ulcer repair material to the tissue in the saliva environment is vital for inhibiting ulcer infection and accelerating ulcer healing. Therefore, we examined the lasting adhesion of PolyLA-Na/PolyLA-based adhesive patches to pig oral musoca tissues in a salivary environment. As shown

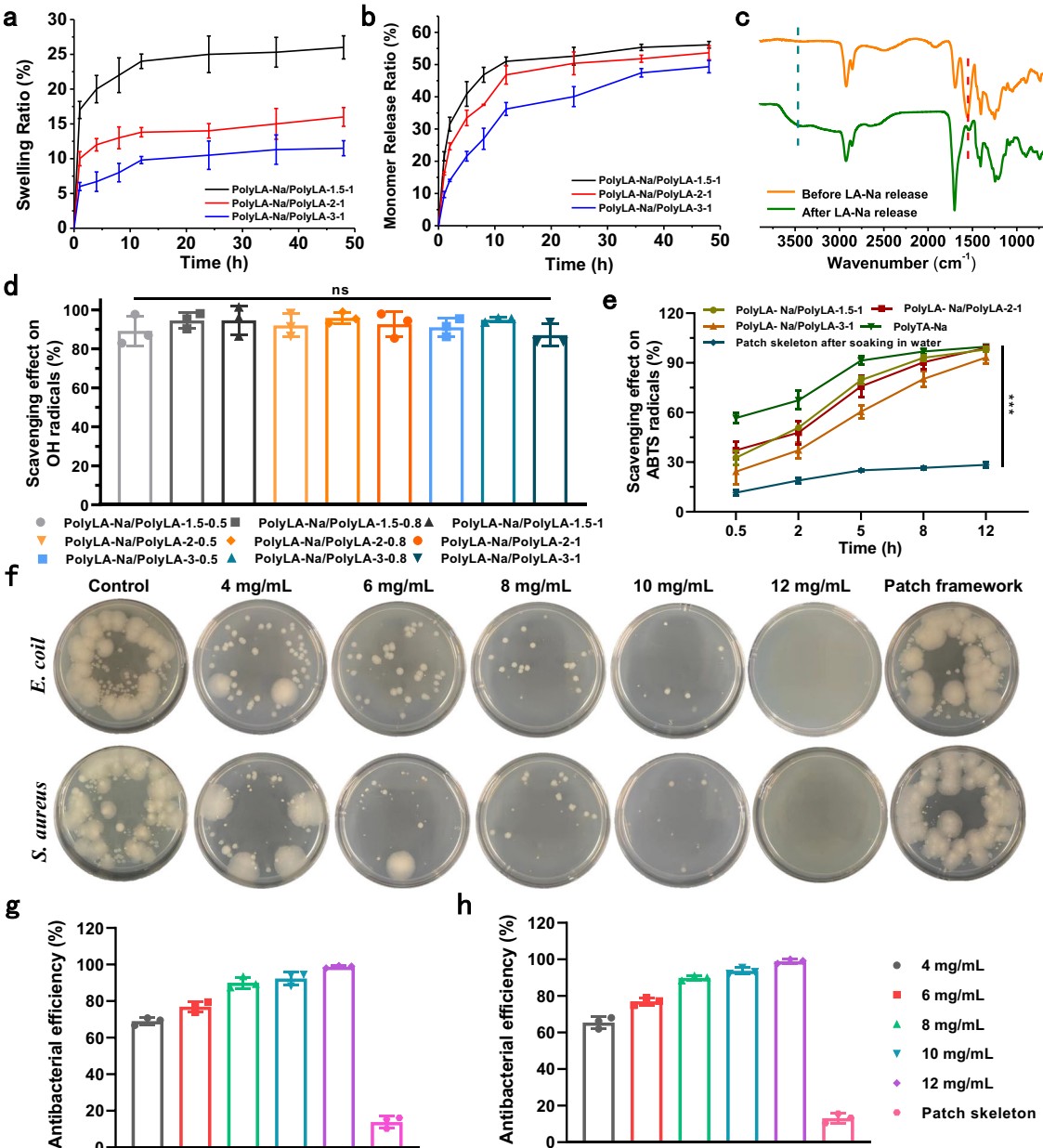

**Fig. 5 | Swelling resistance, LA-Na release, antioxidant and antibacterial properties of PolyLA-Na/PolyLA binary synergistic adhesive patches.** a Swelling behavior of PolyLA-Na/PolyLA adhesive patches with different compositions in artificial saliva ($n = 3$); **b** Monomer release ratio of PolyLA-Na/PolyLA adhesive patches with different compositions in artificial saliva for different times ($n = 3$); **c** FT-IR spectra of PolyLA-Na/PolyLA adhesive patch before and after monomer release; **d** Scavenging effect of PolyLA-Na/PolyLA adhesive patches on OH· radicals at 0.5 h ($n = 3$); **e** ABTS radical clearance ratio of PolyLA-Na/PolyLA adhesive patches

determined at different times ($n = 3$, ***$p < 0.001$); **f** Digital images displaying minimum inhibitory concentration of the PolyLA-Na/PolyLA adhesive patches against *E.coli* and *S.aureus*; **g, h** Sterilization rate of PolyLA-Na/PolyLA adhesive patches with different concentrations against *E.coli* and *S.aureus* at different times ($n = 3$). (All presented data are mean values ± SD from the mean from $n = 3$ independent measurements on independent samples. All $p$ values were calculated using a two-sided Student's $t$ test).

in Supplementary Fig. 17, the adhesive patches can all maintain tight adhesion with oral musoca tissue and can easily lift the tissue even after being soaked and continuously stirred in artificial saliva for 24 h at 37 °C, indicating that the PolyLA-Na/PolyLA-based adhesive patch can achieve durable adhesion to oral musoca tissue against saliva.

### In vitro anti-swelling, antioxidant and antibacterial properties of PolyLA-Na/PolyLA-based adhesive patch

Anti-swelling is critical to patch stability and patient comfort. As shown in Fig. 5a and Supplementary Fig. 18, PolyLA-Na/PolyLA-1.5-0.5 showed the highest swelling rate, which reached 55% after soaking in artificial

saliva for 4 h. With the increase of PolyLA content, the swelling rate of the patch gradually decreased. A reasonable explanation is that the hydrophobic property of PolyLA limited the water adsorption of the patch, and PolyLA broke the orderly arrangement of PolyLA-Na, reduced the water sensitivity of PolyLA-Na network, which also inhibited the swelling of the patch in water. Therefore, the swelling rate of PolyLA-Na/PolyLA-2-1 and PolyLA-Na/PolyLA-3-1 were only 13% and 7% after soaking in artificial saliva for 4 h; with the extension of immersion times, the swelling degree does not increase significantly, indicating that PolyLA-Na/PolyLA-based adhesive patches are stable in saliva environment. Both LA and LA-Na have excellent antioxidant activities

because the two neighboring sulfur atoms in the five-membered ring repel each other at a rather high electron density, leading to the strong reductive property[27]. Therefore, the release of LA/LA-Na can effectively remove ROS from the wound and accelerate wound healing. However, few of the LA-based materials reported so far have harnessed the antioxidant function of LA, since LA monomer is not easily released in an aqueous environment due to its hydrophobic nature. In contrast to this, the dissociation of PolyLA-Na in the aqueous environment is beneficial for the efficient release of LA-Na small molecules; nonetheless, the dissociation rate of the pristine PolyLA-Na film is too fast to satisfy the demand of tissue repair (less than 20 min) (Supplementary Fig. 6). In this work, we anticipate that the introduction of PolyLA can reduce the dissociation rate of the PolyLA-Na and realize the slow and sustainable release of LA-Na by forming multiple -COOH···O=C- hydrogen bonds, while the PolyLA network can maintain long-term adhesion on the wound surface to prevent wound infection. To test our hypothesis, we inspected the monomer release rate of PolyLA-Na/ PolyLA adhesive patches in artificial saliva via UV-vis spectroscopy. As presented in Supplementary Fig. 19, the characteristic absorption peak intensity of the disulfide five-membered ring at 330 nm increases gradually with the immersion time in artificial saliva, and no abrupt release is observed within a short period of time. We calculated the release efficiencies of monomers from PolyLA-Na/PolyLA-based adhesive patches with different compositions (Fig. 5b and Supplementary Fig. 20). As seen from the results, all the adhesive patches can sustainably release antioxidant monomers during the first 12 h in saliva, and attain a release balance after 12 h. This means that the introduction of PolyLA can indeed inhibit the rapid dissociation of the PolyLA-Na network. It is worth noting that the amount of released monomers from all patches is higher than that of LA-Na. A possible reason is that the presence of Na⁺ has a solubilizing effect on LA, so a small amount of LA monomer is also released. From FT-IR spectra of the adhesive patch before and after soaking in artificial saliva for 24 h, we can see that the characteristic absorption peak of -C=O in carboxylates almost completely disappears; while the characteristic absorption peak of -C=O at 1551 cm⁻¹ on carboxyls is preserved. Notably, the -COOH···O=C(OH) hydrogen bond peak appears at around 3500 cm⁻¹ [31], indicating that the PolyLA-Na network is almost completely dissociated, whereas the PolyLA network is retained and stabilized through inter-carboxyl hydrogen bonding cross-links (Fig. 5c). Retention of the PolyLA polymer network is critical for long-term and stable adhesion to tissue to prevent the contamination and infection of the wound site.

Next, the ROS scavenging activity was assayed by determining the antioxidant effect of the PolyLA-NA/PolyLA-based adhesive patch on hydroxyl radicals (OH·), 1,1-diphenyl-2-picrylhydrazyl free radical (DPPH) and 2,2′-azino-bis(3-ethylbenzothiazoline-6-sulfonic acid) diammonium salt radical (ABTS) according to the previously reported method[42,43]. As shown in Fig. 5d, all adhesive patches can achieve more than 90 % OH· scavenging efficiency after 0.5 h incubation. Similarly, the PolyLA-Na/ PolyLA-based adhesive patch also exhibits high DPPH clearance efficiency and can achieve 85 % scavenging activity after 24 h (Supplementary Fig. 21 and Supplementary Fig. 22). The decolorization reaction of DPPH reagent with PBS or PolyLA-Na/PolyLA-based adhesive patches was observed with the naked eye. After 24 h of incubation, the color of DPPH solution turns yellow from dark purple, indicating that the adhesive patches can efficiently scavenge DPPH radicals. Whereas, PBS treatment does not cause color change. The above results intuitively mirrors that the monomer release from the PolyLA-based adhesive patch enhances ROS scavenging activity (Supplementary Fig. 23). In order to prove that the antioxidant property of PolyLA-Na/PolyLA-based patch mainly depends on the LA-Na active small molecules released by PolyLA-Na network in liquid environment, we selected pure PolyLA-Na patch and PolyLA-Na/Poly-2-1 patch skeleton after soaking in water for 48 h as positive control and negative control respectively. The results

show that the pure PolyLA-Na patch has excellent scavenging ability to ABTS radicals, and the scavenging efficiency can reach 90% in 5 h, indicating that PolyLA-Na can quickly dissociate and release LA-Na small molecules in the water environment. Similar to pure PolyLA-Na, PolyLA-Na/PolyLA-based patch also showed satisfactory antioxidant properties, and its scavenging efficiency of ABTS radical increased gradually with the extension of soaking time in water, and the scavenging efficiency reached 75% at 5 h and 99% at 12 h. However, unlike PolyLA-Na and PolyLA-Na/PolyLA-based patches, the PolyLA-Na/PolyLA-2-1 patch skeleton showed negligible antioxidant efficiency, and its scavenging efficiency against ABTS radicals was still less than 30% after 12 h. This confirms that PolyLA-Na network plays a major role in the antioxidant properties of PolyLA-Na/PolyLA-based adhesive patches, while the hydrophobic nature of PolyLA skeleton limits the release of LA monomer in water environment, which mainly plays a role in stabilizing PolyLA-Na network and reducing its dissociation rate in water environment (Fig. 5e).

The warm and humid oral environment, as well as the existence of food debris and secretions, make the oral cavity a breeding ground for various bacteria, which will cause serious infection to the oral ulcer wound[14]. Therefore, the ideal ulcer repair material should also possess a potent antibacterial function. For our PolyLA-Na/PolyLA-based adhesive patch, the released LA-based monomer can kill both Gram-positive bacteria and Gram-negative bacteria by depolarizing the bacterial cell membrane and increasing the permeability of the bacterial cell membrane, resulting in the leakage of substances in the bacteria. In addition, LA-based monomer can also disrupt the metabolism of bacteria to inhibit their proliferation[26,28,29,33,44]. As shown in Supplementary Fig. 24, the PolyLA-Na/PolyLA-based adhesive patches demonstrate a significant inhibition zone on both Gram-position bacteria (S. aureus) and Gram-negative bacteria (E. coil) after co-culturing for 12 and 24 h. Taking PolyLA-Na/PolyLA-2-1 as an example, we investigated the minimum inhibitory concentration (MIC) of the patch by co-culturing it with E.coil and S.aureus for 24 h. As shown in Fig. 5f–h, with the increase of PolyLA-Na/PolyLA-2-1 concentration, its antibacterial efficiency gradually increased, and the MIC of the patch against both E. coil and S. aureus was 12 mg/mL, in which about 6 mg/mL of LA-based small molecules were released, which once again proved that the PolyLA-Na/PolyLA-based patch had a high antibacterial efficiency. Considering its flexibility, durable saliva-resistant adhesion to wet tissue, efficient antioxidative function, and excellent antibacterial bioactivity, we next examined the therapeutic effect of PolyLA-Na/PolyLA-based adhesive patches on oral ulcers.

## In vivo adhesion and biocompatibility of PolyLA-Na/PolyLA adhesive patch

The above outcomes have demonstrated that the PolyLA-Na/PolyLA adhesive patch can achieve immediate adhesion to wet tissue and maintain durable adhesion in a wet environment in vitro. Next, we examined the tissue adhesion of the PolyLA-Na/PolyLA adhesive patch in vivo and its resistance to the dynamic and wet oral environments. Here, we chose PolyLA-Na/PolyLA-2-1 as a representative sample for in vivo experiments in light of its highest wet tissue adhesion, excellent antioxidant and antibacterial activity in vitro. As shown in Fig. 6, the PolyLA-Na/PolyLA-2-1 patch can achieve firm adhesion to the wet tissue surface of the rat buccal mucosa, gingiva, palate mucosa, and tongue (Fig. 6a) and can tolerate the repeated stretching motion of the tongue (Fig. 6b), demonstrating its fine wet and dynamic tissue adhesion. In addition, the durable adhesion of the PolyLA-Na/PolyLA-2-1 adhesive patch to oral mucosa was also tracked and compared with the commercial chitosan oral ulcer repair patch. Remarkably, our PolyLA-Na/ PolyLA-2-1 patch can firmly adhere to the oral mucosa for about 24 h, while the chitosan patch is dissolved within 2 h (Fig. 6c, d). These results indicate that PolyLA-Na/PolyLA patch is promising to be a patch for oral mucosal repair.

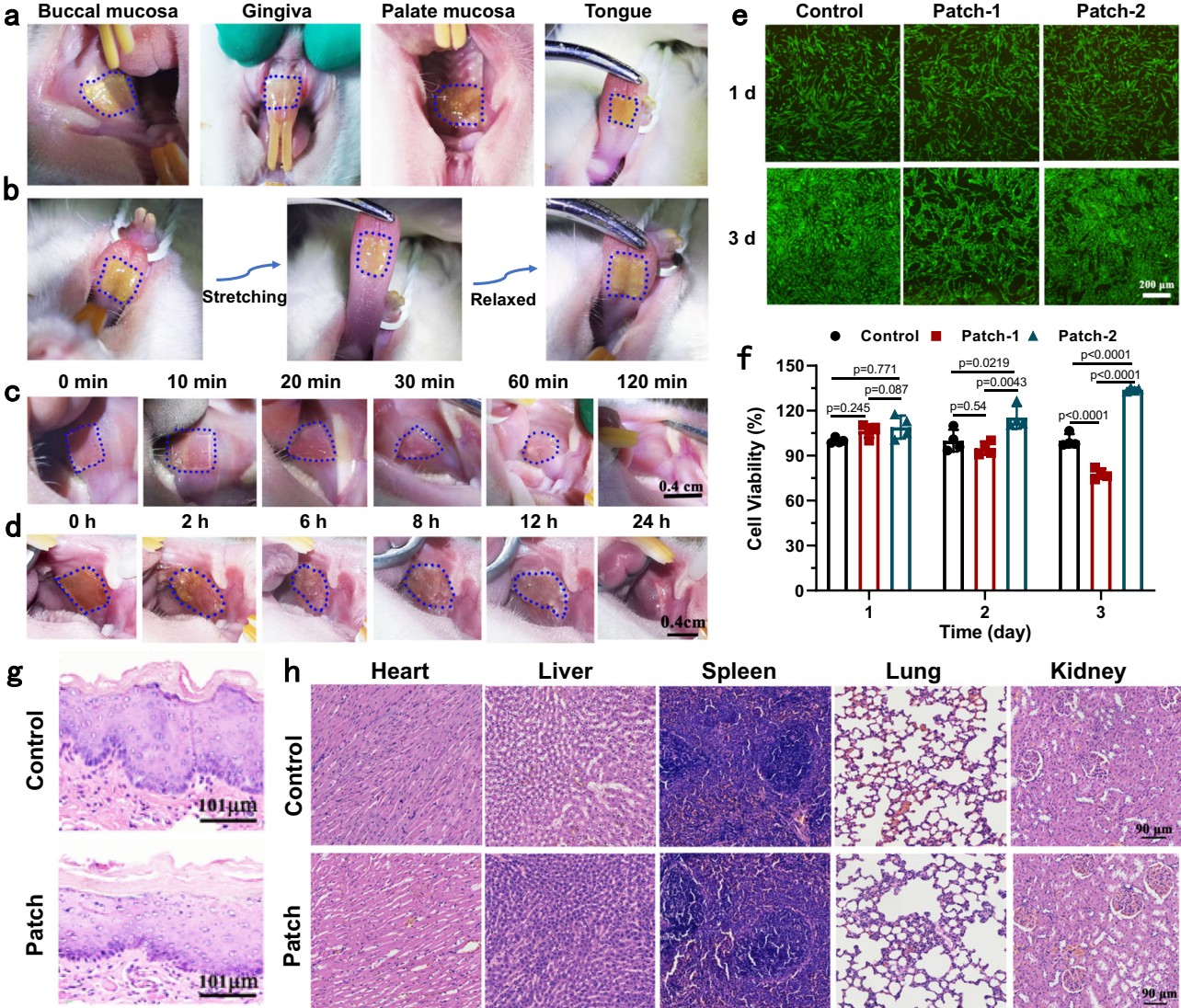

**Fig. 6 | In vivo adhesion and biocompatibility of PolyLA-Na/PolyLA adhesive patch. a** Adhesion of PolyLA-Na/PolyLA patch on the wet surface of the rat buccal mucosa, gingiva, palate mucosa, and tongue; **b** Movement-resistant adhesion of PolyLA-Na/PolyLA-2-1 patch in vivo; **c** Adhesion of chitosan film in oral cavity; **d** Durable adhesion ability of PolyLA-Na/PolyLA-2-1 patch in oral cavity; **e**, **f** Live/Dead staining and cell viability of hGFs cells co-cultured with PolyLA-Na/PolyLA patch for 1 and 3 days ($n = 4$); **g** H&E staining of buccal mucosa tissue after attachment of PolyLA-Na/PolyLA patch for 12 h ($n = 3$ biologically independent samples in each group); **h** H&E staining of main organs after PolyLA-Na/PolyLA patch treatment for 8 days ($n = 3$ biologically independent samples in each group). (All presented data are mean values ± SD from the mean from $n = 4$ independent measurements on independent samples. All $p$ values were calculated using a two-sided Student's $t$ test).

We then assayed the cytocompatibility of the adhesive patches. The cytotoxicity was evaluated by co-culturing human gingival fibroblasts cells (hGFs) and human umbilical vein endothelial cells (HUVECs) (two representative oral mucosal cells in oral mucosal wound healing) with the different mass of PolyLA-Na/PolyLA-2-1 adhesive patch. As shown in Fig. 6e, the hGFs cells spread evenly and survive, which is indicated by observing rich green fluorescent spots in the fluorescent image obtained by live/dead assay in two PolyLA-Na/PolyLA-2-1 patch samples (Patch-1 is 62.5 μg/mL, Patch-2 is 31.25 μg/mL). The cell activity of hGFs cells is quantitatively evaluated by cholecystokinin-8 (CCK-8) assay (Fig. 6f). It is seen that these two adhesive patches show comparable cell viability with the control group after co-culturing with cells for 1 day. Further increasing the culture time to 3 days, the adhesive patch with 62.5 μg/mL content exhibits a slight inhibitory effect on cell proliferation, but the adhesive patch with a content of 31.25 μg/mL demonstrates a conspicuous promotive effect on hGFs cell growth. For HUVECs cells, both patches showed the effect of promoting proliferation (Supplementary Fig. 25). In order to ensure that the PolyLA-Na/PolyLA adhesive patch will not cause cytotoxicity to the organism when used in vivo, we detected the plasma concentration of LA-based molecule in rats at different times by placing the PolyLA-Na/PolyLA-2-1 adhesive patch in the rat oral cavity. The results listed in Supplementary Table 1 show that the highest plasma concentration for the rat is 0.926 μg/mL after PolyLA-Na/PolyLA-2-1 adhesive patch is placed in the rat oral cavity for 6 h, which is much lower than the minimum concentration (31.25 μg/mL) used in vitro cytocompatibility test. These results indicate that the clinically applied dose of PolyLA-Na/PolyLA adhesive patch is cytocompatible. The effective migration of endothelial cells is very important to promote wound healing[17]. Therefore, the impact of the PolyLA-Na/PolyLA patch on HUVECs migration was also investigated by using scratch assays. As shown in Supplementary Fig. 26, the HUVECs migration of Patch 1 and Patch 2 group was faster than that of the control group. Some scratch area was still observed in control group, while a complete closure was

seen in the Patch 1 and Patch 2 group at 24 h. Quantitative analysis displayed that migration rate of Patch 1 and Patch 2 group was 69% and 75% at 12 h and 98% and 99% at 24 h, respectively, which were much higher than those in control group (57% at 12 h, 88% at 24 h). HUVECs tube formation assay was also performed to investigate the pro-angiogenesis potential of the PolyLA-Na/PolyLA patch. As shown in Supplementary Fig. 27, compared with control group, more integrated and diverse well-branched tubes formed in in Patch 1 and Patch 2 group. The number of capillary-like structures in patch 1 and patch 2 was much higher than that in the control group, indicating that the PolyLA-based patch has great potential pro-angiogenesis. The in vitro anti-inflammatory effect of the PolyLA-Na/PolyLA patch was evaluated by co-culturing with LPS stimulated Mouse Monocyte-macrophage Leukemia Cells (RAW264.7). As shown in Supplementary Fig. 28, pro-inflammatory cytokines TNF-a and IL-6 increased significantly after LPS stimulation, but after PolyLA-Na/PolyLA treatment, the production levels of the two pro-inflammatory factors decreased obviously, and the decreasing effect was positively correlated with the patch concentration. These results indicated that the PolyLA-Na/PolyLA adhesive patch exhibited strong anti-inflammatory effect on LPS stimulated RAW264.7 cells. Prior to the therapeutic study, in vivo histocompatibility of the PolyLA-Na/PolyLA adhesive patch was also assessed. The H&E staining results show that compared with the normal tissue, there is no significant inflammation, necrosis, or metaplasia in the buccal mucosa tissue treated with the PolyLA-Na/PolyLA-2-1 patch for 12 h (Fig. 6g), indicating that PolyLA-Na/PolyLA-2-1 patch stimulates no acute reaction for oral mucosa. Histological assessment of major organs (Fig. 6h) including the heart, liver, spleen, lung, and kidney reveals that the PolyLA-Na/PolyLA adhesive patch does not cause any damage to these organs. These data prove that PolyLA-Na/PolyLA adhesive patch does not cause local or systemic toxicity to rats, showing excellent biosafety and biocompatibility.

## Healing efficacy of oral ulcer in the rat model and effect of adhesive patch on the oral microbiota

In order to verify the therapeutic effect of the PolyLA-Na/PolyLA adhesive patch on oral ulcer, we used PolyLA-Na/PolyLA-2-1 patch to repair a 5 mm diameter buccal mucosal ulcer of rats, with the commercial chitosan film-treated groups as positive control and the untreated group as a negative group (Fig. 7a). We observed the change in ulcer size for 8 days and calculated the ulcer recovery efficiency (Fig. 7b, c). The PolyLA-Na/PolyLA patch shows a significant improvement in therapeutic efficiency compared with the commercial chitosan film. On the 8th day, the unhealed area in the PolyLA-Na/PolyLA group is much smaller than that of the chitosan and the untreated group. In particular, the wound healing rate for the group of PolyLA-Na/PolyLA-2-1 on the 4th day can reach $79.37 \pm 5.23\%$, which is more effective than that of the chitosan film group ($62.6 \pm 3.15\%$) and control group ($55.71 \pm 7.97\%$). When the treatment time was prolonged to day 6 and day 8, the therapeutic efficiency of PolyLA-Na/PolyLA-2-1 group was up to $92.90 \pm 1.62\%$ and $99.33 \pm 1.15\%$, respectively, which was still much higher than that of chitosan group ($72.8 \pm 5.63\%$ on 6th day and $88 \pm 1.9\%$ on 8th day) and control group ($66.49 \pm 4.73\%$ on 6th day and $78.43 \pm 4.08\%$ on 8th day). H&E staining and Masson staining were performed to evaluate the ulcer healing and collagen deposition of different treatment groups at the histological level (Fig. 7d, e). On the 4th day, fewer inflammatory cells and smaller ulceration are observed in PolyLA-Na/PolyLA-2-1 group compared with the chitosan group and untreated group. On the 8th day, the ulcers treated with PolyLA-Na/PolyLA-2-1 adhesive patch exhibit a completely regenerated epithelium and well-organized collagen fibers similar to the normal buccal mucosa, whereas there is only partial healing for chitosan film and untreated group. Moreover, immunohistochemistry staining for CK13, which is mainly expressed at the intermediate layer and the parabasal layer[12], shows complete coverage of the epithelium in the PolyLA-Na/PolyLA-2-1 group, whereas this coverage is incomplete in the chitosan group. The worst case is that no epithelium is regenerated in the untreated group (Fig. 7f, h). Next, we further studied changes in buccal mucosa in different groups at the cellular level using an immunofluorescence staining assay. The M1 macrophages (iNOS-positive, iNOS+) and neutrophils (myeloperoxidase (MPO)-positive, MPO+) can produce pro-inflammatory cytokines[21]. We performed immunofluorescence staining of iNOS+ and MPO+ to evaluate the infiltration of inflammatory cells in the ulcer site. After treating the ulcer with the PolyLA-Na/PolyLA-2-1 patch, immunofluorescence staining reveals fewer MPO+ cells and iNOS+ cells compared with the other groups (Fig. 7g, i, j), demonstrating that the PolyLA-Na/PolyLA-2-1 patch treatment effectively prevents the infiltration of inflammatory cells and exhibits a strong anti-inflammatory effect. Previous studies have shown that external stimulation and bacterial infection are the main causes of inflammation[21]. Therefore, in addition to the effective release of LA-Na and LA as well as the lasting adhesion protection of PolyLA-Na/PolyLA patch to ulcer sites, the excellent antibacterial function of the PolyLA-Na/PolyLA patch also plays an important role in reducing inflammation. In order to verify this, we further evaluated the in vivo antibacterial effect of PolyLA-NA/PolyLA patch using a rat oral mucosa ulcer model.

Bacteria at the oral ulcers site were collected by sterilized swabs after the treatment with different methods for 4 days. After 12 h of culture, the bacterial number was evaluated by measurement of $OD_{600}$ nm value. As presented in Supplementary Fig. 29, the $OD_{600}$ nm value of the adhesive patch group is much lower than that of in chitosan film group and untreated group, which means that the PolyLA-Na/PolyLA-2-1 exhibits a strong antibacterial effect in oral ulcer site and is superior to the chitosan film. Oral cavity is one of the diverse microbial habitats in the human body[45]. Maintaining the balance of the oral microbiota is vital to oral and systemic health. There is an increasing number of oral diseases that are considered to be linked to ecological disturbance of oral microbia, such as caries, periodontitis, oral mucosal diseases, and squamous cell carcinoma[46–49]. In addition, dysbiosis of the oral microbiota can also cause opportunistic infections and damage to the mucosal defense system, providing the conditions for viral, spirochetal, fungal, and other infections[50–52]. Therefore, we continued to study the effect of the PolyLA-based patch on the oral microbiota. We sampled the sites of oral ulcers in experimental rats, and compared the characteristics and distribution of oral microflora in each group by 16 s rDNA sequencing. The Venn diagram in Fig. 8a shows the distribution of Amplicon sequence variants (ASV). The normal group and adhesive patch groups share 37 ASVs, while the chitosan film group shares only 16 ASVs with the normal group. Subsequently, the principal component analysis (PCoA) was utilized to evaluate the similarity among samples, and the graph shown is a 95% confidence ellipse (Fig. 8b, c). The only variable in PCoA is the distance between two dots, and the shorter distance means a lower difference. We found that the distance between PolyLA-based patch group and normal group was significantly shorter than that between chitosan film group and normal group, demonstrating a smaller difference between the PolyLA-based patch group and the normal group. Moreover, the 3D PCoA with less statistical loss shows consistent results (Supplementary Movie 1). As shown in the heat map in Fig. 8d, the express extent of consistency in the adhesive patch group is more similar to the normal group than in the chitosan film group. In the chitosan film group, the structure of the flora deviates with respect to the normal group, as exemplified by the overrepresentation of *Fusobacterium* or the underrepresentation of *Helicobacter*. The results of cluster analysis based on the ASVs of the individual samples are shown in Supplementary Fig. 30, with the normal group presenting much less dissimilarity from the PolyLA-based patch group.

In the following experiment, we further analyzed the bacterial species of different groups at the genus level, and found that the dominant bacteria in the normal group are *Muribacter* and

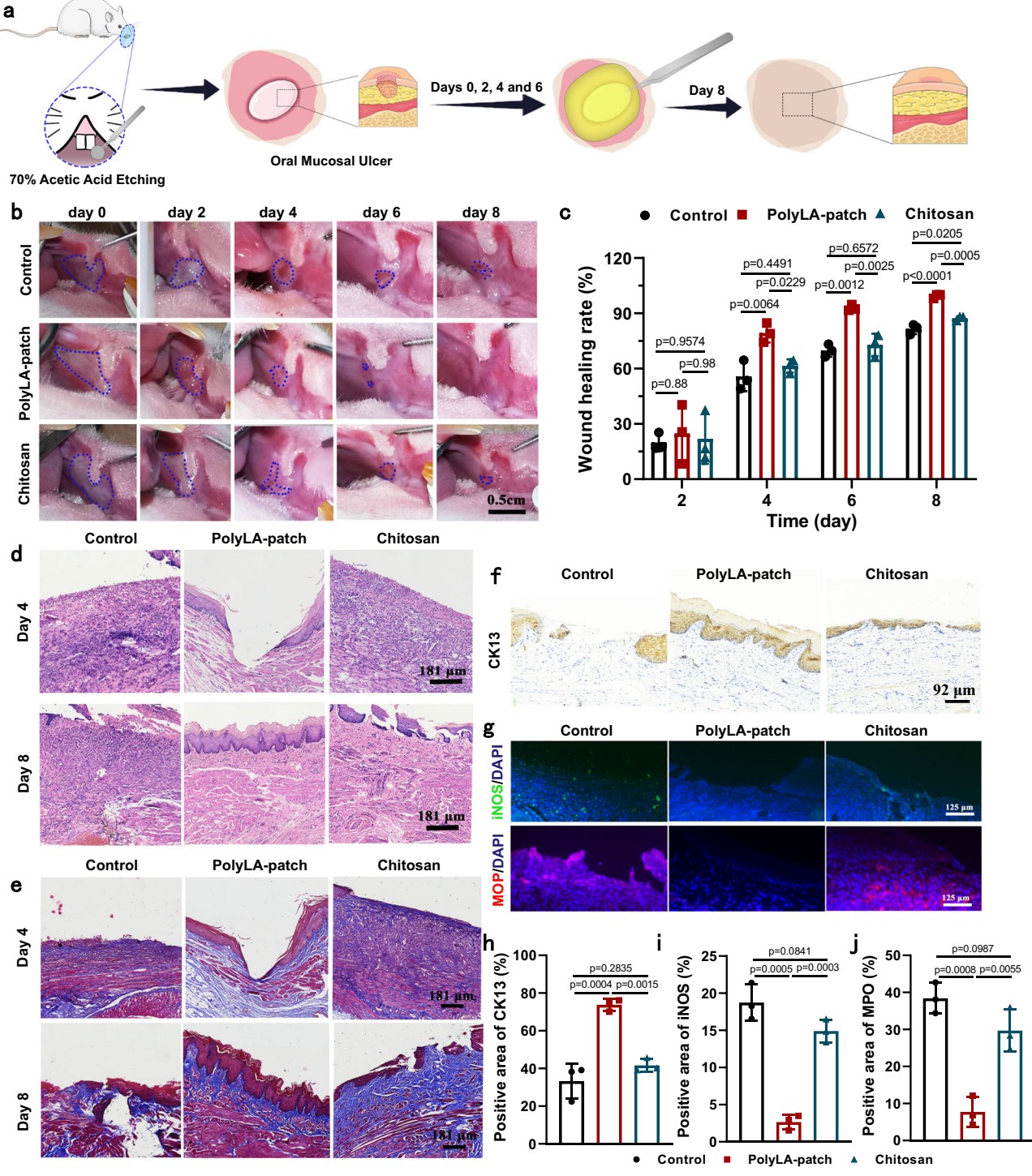

**Fig. 7 | Oral ulcer healing efficacy of PolyLA-Na/PolyLA patch in rat model.**
**a** Scheme depicting creation of rat mucosa defect that was covered by PolyLA-Na/PolyLA patch or chitosan film to assess healing; **b** Photographs of buccal mucosa ulcers in rats treated by PolyLA-Na/PolyLA patch and chitosan film at the 0, 2, 4, 6, and 8 days; **c** Wound healing efficiency of the oral ulcers at days 2, 4, 6, and 8 days after treated with PolyLA-Na/PolyLA patch and chitosan film (*n* = 3); **d, e** H&E staining and Masson's trichrome staining of the regenerated oral mucosa at day 4 and 8 days (n = 3 biologically independent samples in each group);
**f** Immunohistochemistry staining of CK13 antibody in regenerated oral mucosa at 8 days; **g** immunofluorescence staining of iNOS (green) and MPO (red) antibody in regenerated oral mucosa at 4 days; **h–j** Quantification of expression levels of CK13, iNOS, and MPO in regenerated oral mucosa (*n* = 3) CK13 cytokeratin 13, iNOS inducible nitric oxide synthase, MPO myeloperoxidase. (All presented data are mean values ± SD from the mean from *n* = 3 independent measurements on independent samples. All *p* values were calculated using a two-sided Student's *t* test).

*Rodentibacter* (Fig. 8e and Supplementary Fig. 31). After treatment with PolyLA-based patch, there is almost no significant change in the abundance of dominant bacteria in the oral cavity. However, after being treated with chitosan film, the structure of the dominant flora is obviously changed. Among them, the abundance of *Muribacter* is decreased by 50.65%, while the abundance of *Fusobacterium* is substantially elevated. *Fusobacterium* is considered the crucial component to formulate dental plaque, and it is one of the major pathogenic

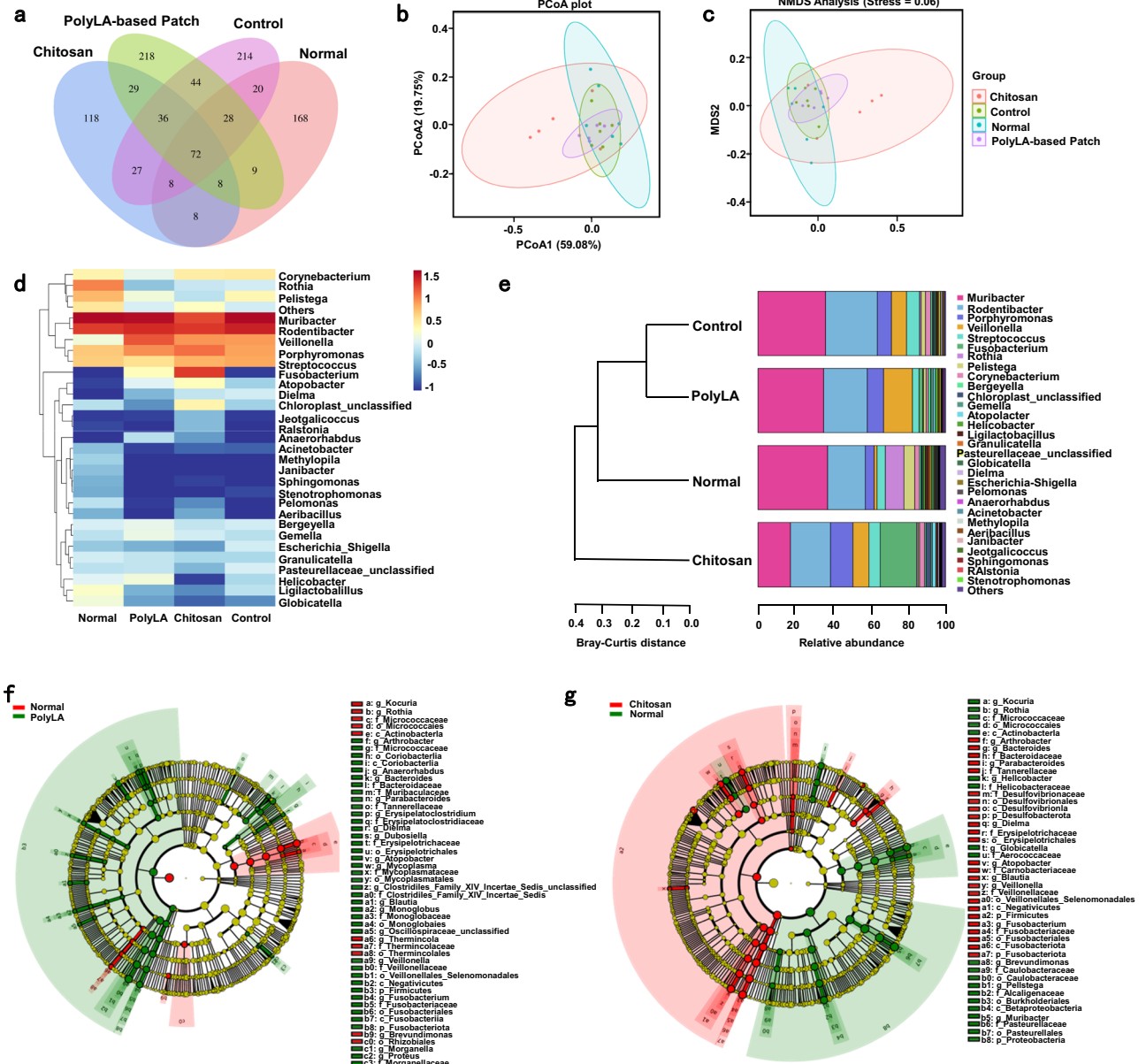

**Fig. 8 | Effects of different treatments on oral microbiota in rats. a** Venn diagram of the oral microbiota in the normal group, control group, chitosan film group, and PolyLA-based patch group; **b,c** PCoA and NMDS analyses of the oral microbiota in the normal group, control group, chitosan film group, and PolyLA-based patch group; **d** Heat map of the oral microbiota in the normal group, control group, chitosan film group and PolyLA-Na/PolyLA patch group; **e** Cluster analysis of the different groups of oral microbiota at the genus level; **f** LEfSe analysis identifying the dominant bacteria in the normal group and PolyLA-based patch group; **g** LEfSe analysis identifying the dominant bacteria in the normal group and chitosan film group.

bacteria in acute necrotizing ulcerative gingivitis[53,54]. Through further bacteria cluster analysis at different levels, we found that the flora structure of the PolyLA-based patch group was more similar to that of the normal group than the chitosan film group at the level of phylum, class, order, family, and species (Supplementary Fig. 32). Linear discriminate analysis Effcet Size (LEfSe) analysis results show that at the level of phylum, there is a significant difference in the dominant bacteria between the chitosan film group and the normal group, but there is no significant difference between the adhesive patch group and the normal group (Fig. 8f, g). These results suggest that the adhesive patch can maintain the stability of the oral microbiota in the process of promoting oral ulcer healing. This has significant implications for the prevention of opportunistic infections. It is worth noting that there is no significant change in oral microbiota in the control group compared with the normal group. A possible reason is that the oral ulcer

model in this work was made by acid etching rather than a bacterial infection. Thus, the microbiota was not significantly changed. But, compared with the group treated with commercial chitosan film, the oral ecology after adhesive patch treatment is more in line with the physicochemical properties of the normal oral cavity. This signifies that chitosan breaks the balance of microbiota.

### Healing efficacy of oral ulcer in a porcine model

Inspired by the positive experimental results of the rat model, we further verified the therapeutic effect of PolyLA-Na/PolyLA-2-1 patch in the mini-pig oral mucosal defect model (8 × 8 mm). The adhesive patch adhered to the defect on the 0th, 3rd and 5th day after oral mucosal injury was induced (Fig. 9a), while the commercial chitosan film was used as a positive group, and the untreated group served as the negative control. On the 3rd, no significant difference in healing effect is

observed among these three groups. On the 5th day, the mucosal defect of the mini-pig was healed to different degrees in all three groups, and the wound closure rates in PolyLA-Na/PolyLA-2-1 (85.41 ± 1.61%) and chitosan film (79.09 ± 4.05%) are significantly higher than the untreated control group (58.16 ± 4.14%) (Fig. 9b–d). On the 8th day, the wound closure rate of the PolyLA-based group can reach 97.8 ± 1.47%, and there are very small and shallow signs of injury observed in the site of mucosal defect, whereas obvious defect is exhibited in the control (83.2 ± 4.4%) and chitosan group (90.4 ± 3.4%) (Fig. 9b–d). The results of H&E staining on the 8th day after treatment with different methods show that completely regenerated epithelium can be seen in PolyLA-Na/PolyLA-2-1 group, while oral mucosa loss and obvious inflammatory cell infiltration are still found in the chitosan group and control group (Fig. 9e). In Masson staining (Fig. 9f), the PolyLA-Na/PolyLA-2-1 group shows a high level of collagen deposition and well-organized collagen fibers; however, the collagen deposition is reduced and the irregular arrangement occurs in the control and chitosan group. Type I and Type III collagens are most commonly expressed in the skin and mucosa tissue[55]. Among them, type I collagen is thick and thought to stabilize the tissue architecture, while type III collagen is thin and mainly contributes to the elasticity of the tissues[56]. Various pathological changes such as wound, inflammation, and fibrosis can induce a change in the expression of collagen in tissues[57,58]. Herein, the Sirius red staining was performed to evaluate the type I and III collagen in different groups. As shown in Fig. 9g, k, compared with the control group and chitosan group, the collagen I/III ratio in adhesive based patch group is significantly reduced due to the high expression of type III collagen, suggesting that the more elastic mucosal tissue is regenerated. Immunofluorescence staining of CK5 also shows that compared with the chitosan group and control group, the adhesive patch group promotes complete epithelial regeneration (Fig. 9h, l).

Blood vessels can supply oxygen and nutrients to the cells associated with healing and are beneficial to the remodeling of the extracellular matrix[59]. The formation of new blood vessels was stained by CD31 with red fluorescence. As depicted in Fig. 9i, m, compared to the control group and chitosan group, more positive staining of CD31 is observed in PolyLA-Na/PolyLA-2-1 group. Quantitatively, the percentage of the positive area of CD31 in PolyLA-Na/PolyLA-2-1 group is 20.03 ± 1.25%, which is significantly higher than that of the control group (5.54 ± 0.92%) and chitosan group (8.73 ± 3.12%), indicating that the adhesive patch facilitates the formation of new blood vessel of the full-thickness oral mucosal defect and the enhancement efficiency is significantly higher than that of chitosan group. Immunofluorescence staining for iNOS shows higher expression of iNOS+ cells in the control group and chitosan group compared with the adhesive patch group, signifying the patch's potent anti-inflammatory effect, which benefits from the continuous release of LA-based active small molecules (Fig. 9j, n). The expression levels of the pro-inflammatory cytokines IL-6 and TNF-α in mini-pig oral mucosa tissue were also evaluated by the immunohistochemical staining. Treatment with PolyLA-Na/PolyLA patch significantly down-regulates the expressions of IL-6 and TNF-α compared with the control group, whereas, the chitosan group only reduces the expression of IL-6, indicating PolyLA-Na/PolyLA patch has a better anti-inflammatory effect (Supplementary Fig. 33). All these results support that the PolyLA-Na/PolyLA-based adhesive patch is superior to the commercial chitosan film in repairing relatively large oral mucosal defects in pigs.

## Discussion

In this study, we reported a self-stabilized and water-responsive deliverable binary synergistic adhesive patch made from coenzyme polymer, poly(α-lipoic acid) (PolyLA) for enhancing mucoadhesion in a wet oral environment, and the same sourced coenzyme salt polymer, poly(sodium α-lipoate) (PolyLA-Na) for accelerating therapy of oral mucosal defect. Compared with other reported PolyLA-based or PolyLA-

Na-based materials[30–32,34–36], the binary PolyLA-Na/PolyLA adhesive patch demonstrates several advantages: (1) self-stabilizing effect of PolyLA network at room temperature without the introduction of potentially cytotoxic double-bond small molecules, metal ions or other energy inputs; (2) slow water sensitivity and excellent extensibility of PolyLA-Na network; and (3) water-triggered continuous release of bioactive LA-based small molecules. Based on the flexible mechanical properties and biological properties of anti-oxidation, anti-inflammation, antibacterial and promoting blood vessels, PolyLA-Na/PolyLA-based patch was developed as an oral mucoadhesive patch to accelerate the repair of oral mucosal injury. Nowadays, various oral mucoadhesive patches have been reported to help accelerate the healing of oral mucosal injury. These patches are often fixed on the surface of oral mucosa by physical adsorption, and endowed with anti-inflammatory function by loading drugs such as Dexamethasone, Ciprofloxacin hydrochloride, Astaxanthin, Oxaliplatin and Mycophenolate[15–17,60–62]. However, only adsorption is often unable to maintain the long-term adhesion of the patch to the wet oral mucosal surface (most of the reported oral mucoadhesion patches adhere to oral mucosa surface in vivo for about 4 h, and a small number of reported mucoadhesive patch can stay on the oral mucosa for 7 h and 10 h), and the loaded drugs may have potential drug toxicity. Although some of these studies have confirmed the effectiveness of patches through volunteers, or clinical trials have been gradually carried out, it is also very important to develop durably adhesive and intrinsically bioactive patches to facilitate the repair of oral mucosal ulcer. In this work, we compared the performance of our PolyLA-Na/PolyLA patch with the reported mucoahesive patches (Supplementary Table 2). We can see that PolyLA-Na/PolyLA adhesive patch offered superior advantages: (1) strong and enduring adhesion in the wet environment of the oral cavity to enable an adequate residence time (>12 h); (2) easy to use; (3) accelerating the repair of oral mucosal injury without loading drugs, growth factors or protein.

In summary, this study presented the strategy for engineering an oral mucosa adhesive patch consisting of PolyLA and PolyLA-Na networks to enhance wet adhesion residence time and strength in a wet environment. Upon applying PolyLA-Na/PolyLA patch onto the mucosa defect tissue, water triggered the sustainable release of bioactive small molecules from the patch, and played an antibacterial, anti-inflammatory, and antioxidant role in regulating the microenvironment of the wound, thus, considerably improving therapeutic efficacy in treating oral mucosa defect. We anticipate that this study will provide a option to develop other tissue adhesives and wound dressings in the future by harnessing naturally occurring bioactive molecules.

## Methods

### Preparation of LA-Na powder, PolyLA, PolyLA-Na, and PolyLA-Na/PolyLA-based adhesive patches

LA-Na powder was obtained by dissolving 3.09 g LA in 300 mL 1.5 mol/L NaOH aqueous solution and then freeze-drying. Dry PolyLA film was made by completely dissolving LA at a concentration of 1.5 mmol/mL in an ethanol solution and then pouring the solution into a mold, which was placed in a constant temperature oven at 37 °C until ethanol was completely volatilized. Similarly, dry PolyLA-Na film was made by completely dissolving LA-Na at a concentration of 1.5 mmol/mL in water solution and then pouring the solution into a mold, which was treated in the same way as above until water was completely volatilized. PolyLA-Na/PolyLA-based adhesive patches were prepared through two steps. First, different concentrations of LA ethanol solutions and 1.5 mmol/mL LA-Na aqueous solution were separately prepared and mixed with different volume ratios, and stirred thoroughly until the mixture became a homogeneous solution. Subsequently, the mixed polymer solution was poured into the mold, which was then placed in a constant temperature oven at 37 °C until the mixed solvent of ethanol and water was completely volatilized.

## Molecular dynamics simulation

To compare the structural stability of PolyLA, PolyLA-Na, and PolyLA-Na/PolyLA binary system, three periodic model cells, PolyLA, PolyLA-Na, and PolyLA-Na/PolyLA-2-1 were built. In each case, at the beginning

of the simulation, the energy of the model system was minimized. After that, a molecular dynamic of 500 ps at a constant temperature (300 K) and pressure (1 atm) was performed, which brought the system into a reasonable preequilibrated configuration. Then, a further 1000 ps NVT

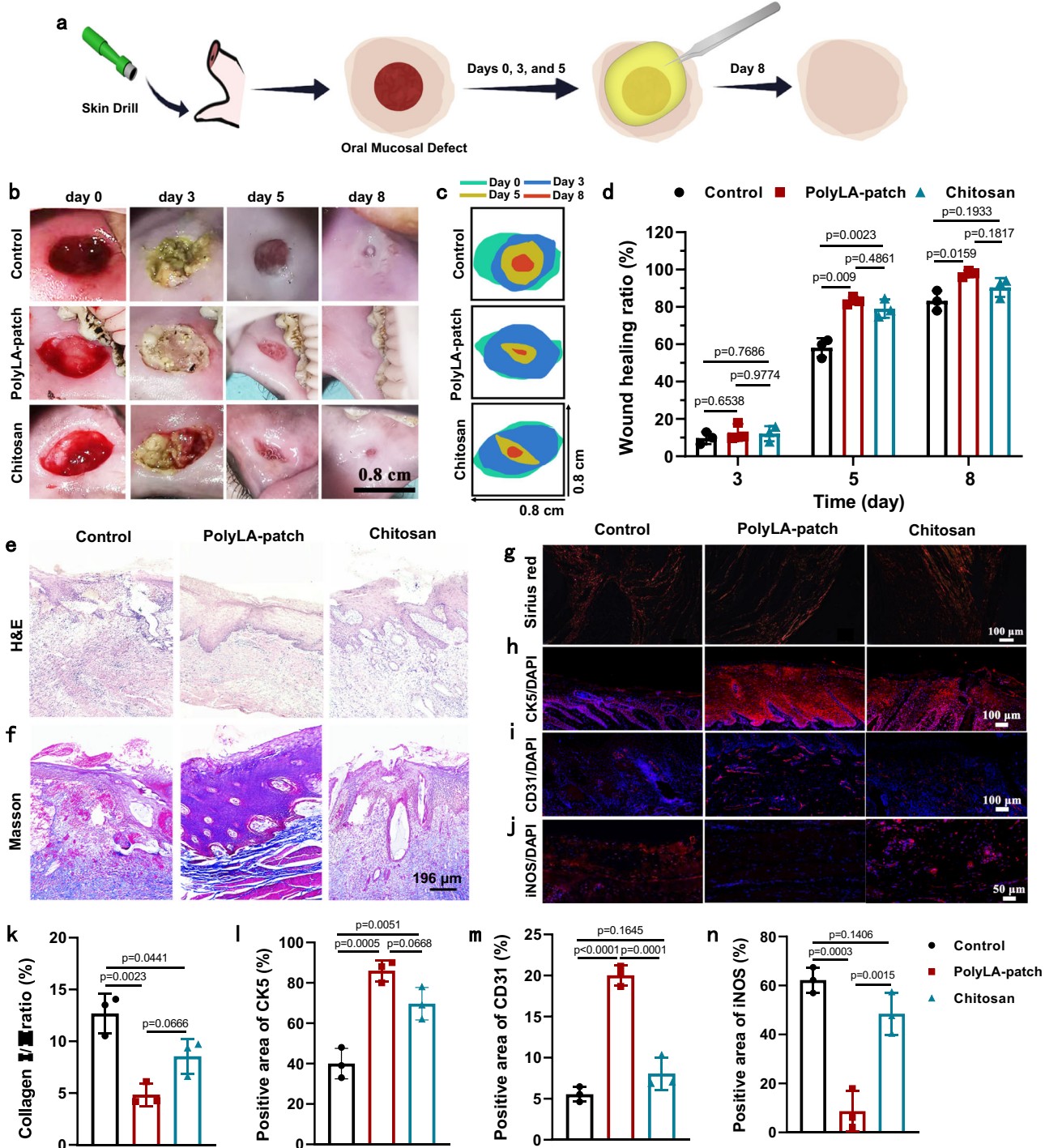

**Fig. 9 | Oral ulcer healing efficacy of PolyLA-Na/PolyLA patch in mini-pig model.** **a** Scheme depicting creation of mini-pig mucosa defect that was covered by PolyLA-Na/PolyLA patch to accelerate healing; **b** Photographs of buccal mucosa defect in *mini*-pig treated by PolyLA-based patch and chitosan film at the 0, 3, 5, and 8 days; **c** Tracking of wound-bed closure after treating with different methods for 0, 3, 5, and 8 days; **d** Wound healing efficiency of the oral ulcers at days 3, 5, and 8 days after treated with PolyLA-Na/PolyLA patch and chitosan film (*n* = 3); **e**, **f** H&E staining and Masson's trichrome staining of the regenerated mini-ping oral mucosa at day 8 (*n* = 3 biologically independent samples in each group); **g** Sirius red staining

of type I collagen (red) and type III collagen (green) in the regenerated mini-ping oral mucosa at day 8; **h–j** Immunofluorescence staining of CK5 (red), CD31 (red) and iNOS (red) antibody in regenerated oral mucosa at 8 days; **k** Analysis of the collagen I/III ratio in Sirius red staining (*n* = 3); **l–n** Quantification of expression levels of CK5, CD31, and iNOS in regenerated oral mucosa (*n* = 3) CK5 cytokeratin 5, CD31 platelet endothelial cell adhesion molecule-1, iNOS inducible nitric oxide synthase. (All presented data are mean values ± SD from the mean from *n* = 3 independent measurements on independent samples. All *p* values were calculated using a two-sided Student's *t* test).

ensemble molecular dynamics simulation was conducted at 300 K to track changes in the system. In the simulation, Packmol[63] was used to build the initial configuration of all the model systems. Large-scale Atomic/Molecular Massively Parallel Simulator[64] and the polymer consistent force field[65] were used to perform the molecular simulations. The time step was fixed at 1.0 fs, and the temperature and pressure were controlled by the Nosé-Hoover thermostat-barostat[66]. A van der Waals interaction cutoff of 1.5 nm was employed, and the Particle-Particle-Particle-Mesh (PPPM) method was used to account for the long-range electrostatic interactions. The atomic coordinates were saved every 1 ps for further analysis.

## Measurement of mechanical properties of PolyLA-Na/PolyLA-based adhesive patches

The mechanical properties of patches at room temperature were tested on an Instron 2344 Microtester in an air environment. At least four specimens were tested for each property. For the tensile test, all samples were cut into rectangular pieces with a dimension of 30 mm × 8 mm × 0.5 mm and the extension rate was set at 50 mm min$^{-1}$.

## Measurement of adhesion strength of PolyLA-Na/PolyLA-based adhesive patches

Lap-shear testing was performed on the above electromechanical tester to measure the adhesive strength of the PolyLA-Na/PolyLA-based adhesive patches with a strain rate of 50 mm min$^{-1}$. The oral mucosa of pigs was used as representative tissue for adhesion strength test, and the oral mucosa was purchased from the local slaughterhouse. Prior to adhesion, pig oral mucosa tissue was soaked in artificial saliva for 30 min. Subsequently, the patch was applied to the surface of oral musoca tissue, and the adhesion area was recorded. Adhesion force was measured immediately after contacting the patch with the pig oral mucosa tissue, and the adhesion strength was calculated using the following equation: adhesion strength = $F_{max}$/S, where $F_{max}$ is the maximum adhesion force during the lap-shear test, S is the adhesion area. All tests were repeated at least four times.

The durable adhesion of PolyLA-Na/PolyLA-2-1 patch to oral mucosa tissue in wet environment was investigated by placing PolyLA-Na/PolyLA-2-1 patch adhered-oral mucosa in a beaker filled with artificial saliva and adjusting the speed of magnetons in the beaker to 120 r/min. After stirring continuously at 37 °C for 24 h, the adhesion of PolyLA-Na/PolyLA-2-1 patch to oral mucosa was observed.

## Monomer release behavior of PolyLA-Na/PolyLA-based adhesive patch in artificial saliva

Firstly, the content of the PolyLA-Na/PolyLA patches with different composition was fixed to 10 mg/mL in artificial saliva, and then these samples were placed in a 37 °C constant temperature incubator. The absorbance of the supernatant at 330 nm wavelength was measured by a UV−visible spectrophotometer (Genesys 180, Thermo Scientifc) at different time points. The release amount of small molecular monomer is calculated by standard curve equation, and the release rate is the ratio of the released small molecular mass to the total mass of the PolyLA-Na/PolyLA patch in artificial saliva.

## ROS scavenging ability of PolyLA-Na/PolyLA-based adhesive patches in vitro

The scavenging effect of pure PolyLA-Na, PolyLA-Na/PolyLA-based adhesive patches with different compositions, and PolyLA-Na/PolyLA-2-1 patch skeleton after soaking in water for 48 h on ABTS radicals were tested by following the instructions of the ABTS free radical scavenging efficiency kit (Solarbio, Beijing, China).

Then, the scavenging activity of PolyLA-Na/PolyLA-based adhesive patch with different compositions on DPPH was evaluated according to the previously published protocol[42]. First, DPPH was dissolved in 80% ethanol to form a 0.01 mmol/L solution. Then 10 mg

of the patch was immersed in 8 mL DPPH solution for different times. The absorbance of the supernatant was recorded at 520 nm wavelength on a UV−visible spectrophotometer (Genesys 180, Thermo Scientifc) at different mixing time points. In the same way, 10 µL PBS was added into the DPPH solution as the control group. The scavenging ability of the PolyLA-Na/PolyLA patch was assayed by the following equation: Scavenging ability (%) = 1 − $A_{sample}$/$A_{control}$ × 100%. Where $A_{sample}$ and $A_{control}$ denote the absorbances of the sample and control group. All tests were repeated thrice.

The Fenton reaction was utilized to evaluate the scavenging effect of different PolyLA-Na/PolyLA-based adhesive patches on hydroxyl radicals. 10 mg patch sample was incubated with 600 µL FeSO$_4$ solution (2 mmol/L) and 500 µL safranin O solution (360 µg/mL) for 10 min. Then, 800 µL of H$_2$O$_2$ solution (6 wt%) was added to the above solution, followed by further incubation in an oven at 55 °C for 30 min. For comparison, the blank group was prepared by adding the same volume of deionized water (DIW) to replace the adhesive patches without changing other condition. To prepare the control group, the patch and H$_2$O$_2$ solution were respectively replaced with 10 and 800 µL of DIW to form a mixture. At the prescribed time, the mixture was removed from the oven and cooled down to room temperature. Finally, the absorbance of the supernatant was measured at 492 nm on a microplate reader (Infinte M200 PRO, Tecan, Switzerland). The scavenging ability of hydroxyl radicals was calculated by the following equation:

Scavenging ability (%) = 1 − ($A_{sample}$ − $A_{blank}$)/($A_{control}$ − $A_{blank}$) × 100%. Where $A_{sample}$, $A_{control}$, and $A_{blank}$ denote the absorbances of the sample, control, and blank group. All tests were repeated thrice.

## Antibactrial ability of PolyLA-Na/PolyLA-based adhesive patches in vitro

The inhibition zone of PolyLA-Na/PolyLA-based patch was tested by co-culturing the patch with bacteria. *S. aureus* (gram-positive aerobic) and *E. coil* (gram-negative aerobic) were selected as model bacteria in this study. *S. aureus* and *E. coil* were cultured in sterilized LB liquid medium at 37 °C overnight in an incubator shaker. The bacterial suspension was removed after 12 h incubation and diluted with the corresponding medium until the optical density (OD, 600 nm) reached -0.01 (the concentration of bacteria is about 10$^7$ CUF m L$^{-1}$). Then, 50 µL of *S. aureus* and *E. coil* were evenly dispersed on a LB/agar plate; later, the PolyLA-Na/PolyLA based patch with a diameter of 7 mm was placed to the agar medium and incubated at 37 °C for 12 and 24 h. After incubation, the zones of inhibition were observed.

The MIC of PolyLA-Na/PolyLA-based patch against both *E. coil* and *S. aureus* was assessed by the similar procedures described in previous work[67]. First, the patch was weighed to obtain 4−12 mg of samples. Each sample was placed in one sterilized centrifuge tube. Second, based on the optical density (at 600 nm, OD600) of the bacterial suspension in LB medium (i.e., a bacterial suspension with an OD of 1 indicates the concentration of 1 × 10$^9$ CFU/mL), the cultured bacterial suspension was diluted to 1 × 10$^4$ CFU/mL by LB medium. Next, 1 mL of diluted bacterial suspension and pure LB medium (background group) was added with the patch sample (to achieve the final concentrations from 4−12 mg/mL) respectively, and incubated at 37 °C for 24 h by a constant temperature shaker. Also, 1 mL of diluted bacterial suspension was incubated in one sterilized centrifuge tube (at 37 °C for 24 h) and used as the control group (untreated bacteria). Next, the values of different groups in OD600 were recorded. The bacterial inhibition efficiency was calculated using the following equation: Antibacterial efficiency (%) = 1 − ($OD_{sample}$ − $OD_{backgroud}$)/($OD_{control}$ − $OD_{backgroud}$) × 100%. $OD_{smaple}$, $OD_{control}$, and $OD_{backgroud}$ were the OD600 values of the sample, control and background group. All tests were repeated three times. The MIC of patch was defined as the MIC for antibacterial efficiency >99%. Finally, the bacterial suspension of the patch-treated or control group was diluted by PBS by 10,000 times and spread

evenly on an agar plate/medium, and incubated at 37 °C for 24 h to observe the colony growth.

## In vitro cytotoxicity assay

The primary hGFs were isolated from adolescent gingival tissues obtained from the alveolar surgery at the Department of Periodontology, the Affiliated Hospital of Qingdao University. The gingival tissue was washed five times with phosphate buffered saline (PBS) and cut into 1 mm thick pieces. The gingival tissue pieces were digested with 2.5 g/L trypsase and 2 g/L type I collagenase for 1 h at 37 °C and centrifugated at $300 \times g$ for 5 min. The cell pellet was washed three times with Dulbecco's modified Eagle medium (DMEM, Biological Industries, Israel) and resuspended with complete medium (DMEM supplemented with 1% penicillin/streptomycin and 10% fetal bovine serum) (FBS, Biological Industries, Israel) and cultured at 37 °C with 5% $CO_2$ in the air. The culture medium was replaced every 3 days and when the cells grew to 80% the following experiments were conducted. The present study with patients' informed consent was approved by the Ethics Committee of the Affiliated Hospital of Qingdao University, China. The human umbilical vein endothelial cells (HUVECs, iCell-h110) were purchased from Shanghai Institutes for Biological Sciences, China. The extraction of PolyLA-Na/PolyLA-2-1 was prepared by immersing 10 mg PolyLA-Na/PolyLA-2-1 into 10 mL of complete growth medium-Dulbecco's modified Eagle medium (DMEM for hGFs cells, Biological Industries, Israel) or Endothelial Cell Medium (ECM for HUVECs cells, Sciencell 1001) supplemented with 1% penicillin/streptomycin and 10% fetal bovine serum (FBS, Biological Industries, Israel) and placed in 37 °C for 24 h. Firstly, the primary hGFs (4000 cells/well) and HUVECs (5000 cells/well) were seeded into 96-well plate, and incubated for 24 h at 37 °C, and then the culture medium was removed and PolyLA-Na/PolyLA-2-1 extract was added to each well. After culturing for 1, 2, and 3 days, 10 µL of the cell counting kit-8 reagent (CCK-8, Absin Bioscience Inc., China) was added into each well and incubated in dark at 37 °C for 1 h, the absorbance value at 450 nm was measured using Microplate Reader (SynergyH1/H1M, Bio-Tek, China). At the same time, the cells were stained with Live and Dead Stain Kit for 15 min and imaged by fluorescence microscope (Nikon, Japan).

## Cell migration assay

HUVECs were incubated in 12-well plates with ECM. After full confluence, a line of HUVECs was gently scratched first by using a 200-µL pipette tip and washed with PBS for three times. Then, ECM with 2% FBS and extraction of PolyLA-based patch made by ECM with 2% FBS was added. The cell migration was captured at regular intervals using an inverted microscope (Nikon, Japan) after 0, 6, 12, and 24 h of incubation. Migration rate (%) was quantified using ImageJ software.

## Tube formation assay

The bottom of 24-well plates was coated with 280 µL of 2:1 diluted Matrigel (ABMbio, Shanghai, China) and then incubated at 37 °C for 30 min. HUVECs (30,000 cells/well) were gently seeded on the Matrigel and cultured in ECM and extraction of PolyLA-based patch made by ECM for 4 h. The cells were stained with Calcein-AM for 15 min and capillary-like structure was imaged by fluorescence microscope (Nikon, Japan).

## In vitro anti-inflammation assay

RAW264.7 cells (RAW264.7 cells (CL-0190), purchased from Procell Life Science&Technology Co., Ltd. RAW264.7 cells were derived from Abelson murine leukemia virus-induced tumors of adult male mice) were seeded in 12-well plates ($2 \times 10^5$ cell/well) and cultured in DMEM with 10% FBS for 24 h. The culture medium was replaced with 500 µL of fresh culture medium, culture medium with 10 µg/mL lipopolysaccharide (LPS), extractions of PolyLA-based patch with 10 ng/mL LPS and co-cultured for 24 h. The cell supernatant was collected and

centrifuged at 300 g for 20 min and the concentration of TNF-α and IL-6 in cell supernatant was measured by enzyme-linked immunosorbent assay (ELISA) Kit according to the manufacturers' instructions (Boster, Wuhan, China).

## Establishment and treatment of oral ulcer model in Rats

All animal experiments were conducted with the permission of the Institutional Animal Care and Use Committee of the Qingdao Medical College of Qingdao University, China (Approval NO. QDU-AEC-2023269). Sprague Dawley rats (27 male, 8-week-old, 250 g of averaged body weight) were anesthetized by intraperitoneal injection of 2% pentobarbital sodium (400 µL/g) (sex does not have a direct effect on the disease we studied, so it was not considered in the design and analysis). The oral ulcer model was induced by placing a round filter paper (5 × 5 mm) soaked with 70% acetic acid on the buccal mucosa of rats for 3 min, and a mucosal discoloration was observed immediately after acid stimulation. At 2 days post formation of oral ulcer (day 0), the rats were randomly divided into three groups. In the experimental group, a 5 × 5 mm size of PolyLA-Na/PolyLA-2-1 patch was attached to the oral ulcer site. In the positive control group, a commercial chitosan film was applied to the mucosa ulcer site. No treatment rats with oral ulcer were used as the negative control group. The animals were treated with the patch at 2, 4, and 6 days after the first dressing (0 d), and gross observation was photographed before application of each material, and the oral ulcer wound closure rate was analyzed by ImageJ.

## Establishment and treatment of mucosa defect model in mini-pig

Five female mini-pigs (5–6 kg, 6-month-old) fasted for 12 h before surgery (sex does not have a direct effect on the disease we studied, so it was not considered in the design and analysis). Firstly, the animal was anesthetized with intramuscular injection of Shutai 50 (10 mg/kg) and propofol (5 mg/kg) by intravenous injection. Then, an 8 mm × 8 mm of mucosa defect was made with a skin drill at the buccal mucosa of minipigs. After hemostasis, the defect site was covered with PolyLA-Na/PolyLA-2-1 patch, while the chitosan film was used as the positive control. The size of the PolyLA-Na/PolyLA-2-1 patch and chitosan film are all disks in diameter 1.2 cm. The healing of mucosa defect was photographed at 3 and 5 days after treating with PolyLA-Na/PolyLA-2-1 patch and chitosan film. The wound closure rate was analyzed by ImageJ.

## Histological and immunofluorescent staining

The rats were sacrificed on 4 and 8 days after treatment, and the pigs were sacrificed 8 days after surgery. The buccal mucosa around the ulcer was collected. For histological analysis, the collected tissues were fixed in formalin (Boster Biological Technology Co., Ltd), embedded in paraffin and stained with hematoxylin and eosin (H&E, Solarbio, China) and Masson's trichrome (Solarbio, China). For immunofluorescence analysis, immunofluorescence staining of iNOS and MPO was performed to evaluate the percentage of inflammatory cells; CK5 and CK13 stainings were performed to evaluate the regeneration of the epithelium; CD31 staining was done to test the formation of new blood vessels; Sirius red is specific staining for the collagen; IL-6 and TNF is staining for inflammatory factors. To evaluate the in vivo biosafety, the major organs of the heart, liver, spleen, lung, and kidney of rats were harvested on the 8th days after treatment and stained with H&E. Three fields of view per section were analyzed for iNOS and MPO staining, and one field of view per section was analyzed for CK5, CK13, CD31 and Sirius red staining, and all results were analyzed by a pathologist who were blinded to experimental design.

## Evaluation of the effect of PolyLA-Na/PolyLA patch on the oral microflora

The site of oral ulcer in the normal group, control group, PolyLA-Na/PolyLA patch group, and chitosan film group were swabbed ten times

with a sterile cotton swab. The analysis of the bacterial community was sequenced on the Illumina NovaSeq platform of Hangzhou Lianchuan Biologicals, Zhejiang, China.

## Statistical analysis

Data were presented as means ± standard deviations. Data from experiments were analyzed with GraphPad Prism 8.0. One-way analysis of variance (ANOVA) followed by Tukey's post hoc test was used for comparisons among multiple groups in the vitro experiments, and the student's test was used to determine the differences between two groups of data in the cell and animal experiments.

## Reporting summary

Further information on research design is available in the Nature Portfolio Reporting Summary linked to this article.

## Data availability

The data generated in this study are provided in the Supplementary Information/Source Data file. The full image dataset is available from the corresponding author upon request. Source data are provided with this paper.

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

## Acknowledgements

The authors gratefully acknowledge the support for this work from the National Natural Science Foundation of China (Grant No. 52233008 and 51733006, 52303201, Grant recipient: W.L., C.C.), Program for talents in Scientific and technological Innovation of Tianjin University (Grant No. 2023XQM-0047, Grant recipient: C.C.), the Innovation and technology program for the excellent youth scholars of higher education of Shandong province (Grant No. 2019KJE015, Grant recipient: Q.Z.), Traditional Chinese Medicine Science and Technology Project of Shandong province (Grant No. 2021Q069, Grant recipient: Q.Z.), the Leading Project of Science and Technology of Yantai Development Zone (Grant No. 2021RC016, Grant recipient: Q.Z.), Young Taishan Scholars Program of Shandong Province (Grant No. tsqn202306272, Grant recipient: Q.Z.), Innovation and technology program for the excellent youth scholars of higher education of Shandong province (Grant No. 2019KJE015, Grant recipient: Q.Z.).

## Author contributions

W.L., Q.Z., and C.C. conceived the project and designed the experiments. C.C. synthesized adhesives, performed the sample fabrication, carried out the mechanical and adhesion experiments, and analyzed the data. L.M. and D.W. conducted all animal experiments and analyzed the data. P.J. contributed to oral microbiota test and analyzed the data. C.C., L.M., and P.J. wrote the manuscript. W.L. and Q.Z. revised the draft. All authors read and approved the final manuscript.

## Competing interests

The authors declare no competing interests.
