## [Peer Review File · Nature Communications]

Reviewers' Comments:

Reviewer #1:

Remarks to the Author:

This manuscript develops a hydrogel-based tissue adhesive for prolonged, sustainable oral ulcer treatment. Adopting hydrogel-based tissue adhesive to treat oral ulcer is novel to me. The overall material characterizations, in vitro demonstrations, and animal studies are comprehensive and effective. However, the key concern raised by the reviewer is that the intellectual merit of this work is limited. It is not clear why PolyLA-Na/PolyLA tissue adhesive outperforms many other tissue adhesives intensively studied in recent decades. The reviewer lists other comments in detail below.

1. The authors systematically compare PolyLA-Na/PolyLA, PolyLA, and PolyLA-Na tissue adhesives. However, there is little discussion on how the tissue adhesive compared with many other tissue adhesives recently studied (H. Yuk et al., *Nature*, 575, 2019; H. Yuk et al., *Nature Biomedical Engineering*, 5, 2021; J. Li et al., *Science*, 357, 2017). The reported adhesion strength in these literature is about 100 kPa, much larger than this work.
2. The intellectual merit of the material system is limited. The key design of this work introduces hydrogen bonds between PolyLA-Na and PolyLA to enable a prolonged degradation, resulting in sustainable drug delivery. This design is obvious to the reviewer.
3. In Line 99 on Page 5, "... the amorphous PolyLA serves to reduce the crystallinity of PolyLA-Na, thus increasing the elasticity and extensivity of the PolyLA-Na film." To clarify this physical picture, the authors are recommended to provide stress-strain curves of materials in Fig. 2. The y-axis in Fig. 2g-2i should be strength instead of stress. The y-axis in Fig. 2j-2l should be ultimate strain instead of strain.
4. In Line 106 on Page 5, "...the adhesive PolyLA hydrophobic network is still maintained...", which is unclear. What is the crosslinker for the PolyLA hydrophobic network?
5. The authors discussed the crystalline-to-amorphous transition. To quantify the crystallinity, the authors are recommended to perform differential scanning calorimetry to measure the crystallinity in the series of samples directly.
6. Some fonts in the main figures are too small to be visualized. The authors are suggested to revise the fonts in Fig 2, 3, 4, 7. The data presented in Fig. 4 is too crowded. The authors can keep the most critical data in Fig. 4 while leaving detailed raw data in supplementary materials. The fonts in Fig. 7f and 7g are too small.

Reviewer #2:

This study, the authors presented the strategy for preparing an oral mucosa adhesive patch consisting of PolyLA and PolyLA-Na networks. The binary synergistic PolyLA-Na/PolyLA patch showed durable adhesion to oral mucosal wound and water-responsive sustainable release of bioactive small molecules. In treating oral mucosa defect, the adhesive patch showed considerably improved therapeutic efficacy. This paper is well-organized with systematic experimental data and logical language description. I recommend it to be published in the journal of Nature Communications after the following issues be addressed:

1. Measurement of adhesion strength of PolyLA-Na/PolyLA adhesive patches to skin tissue, does the sample thickness affect the adhesion strength/adhesion force? Additionally, the source of the skin tissue should be specified.
2. The method section shows that the animals were treated with the patch every two days after the first dressing, but the authors only displayed that the patches could maintain stable adhesion with skin tissue after being soaked in artificial saliva (Supplementary Fig. 16) and with oral mucosa of rat (Fig. 5d) for 24 hours, how about the patch state for 48 hours. How long can the patch adhere stably to oral mucosa?
3. The adhesive strength of the PolyLA-247 Na/PolyLA adhesive patch to the artificial saliva-treated skin tissue was tested before swelling of the patch. The adhesion strength of the patch after swelling is suggested to test and discuss, as the oral environment is aquiferous.
4. Does the monomer release affect the adhesive strength of the PolyLA-247 Na/PolyLA adhesive patch. The adhesive strength of the patch after soaking in artificial saliva for different times (e.g. 12, 24, 36, 48 h) is suggested to measured.
5. In treatment of mucosa defect model in mini-pig, the sizes of the PolyLA-Na/PolyLA-2-1 patch and chitosan film should be specified?

Reviewer #3:

Remarks to the Author:

A. Key results

This interesting study by Cui et al describes the production of a polymer mucoadhesive patch based on PolyLA-Na and PolyLA that provides the basis for the release of poly alpha-lipoic acid, which the authors suggest, has a range of diverse biological activity including antimicrobial, angiogenic whilst also regulating the immune response and increasing wound healing. These mucoadhesive patches were tested on experimentally induced oral ulcers/wounds in mice and pigs to show efficacy in vivo.

B. Significance

There are now many reports going back several years on of the use of fabricated polymer-based mucoadhesive patches based on gels, electrospun fibers etc. for the treatment of oral ulcers or other oral lesions. These reports differ on the polymers used, type of fabrication process and therapeutic drug/protein delivered (Colley et al Biomaterials 2018, 178:134-146; Teno et al (2022) J. Funct. Biomater. 13, 170; Paris et al. Acta Biomaterialia 2021, 128, 222-35; Liu et al, Small, 2022, 18, 2201561, Zhang et al, 2022 Materials & Design 223, 111131, Teno et al 2022 Functional Biomaterials 13:170, Alves 2020, Pharmaceutics 12:657). There are reports of these progressing to the clinic, completing phase 2 clinical trials (Ibrahim et al. BMC Oral Health (2023) 23:99; Brennan et al 2021 DOI: 10.1111/jop.13270). Moreover, some reports have showed that the polymers themselves may also have biological effects (Edmans 2022, doi.org/10.1016/j.jconrel.2022.08.016). This manuscript adds to the increasing list of mucoadhesive polymers and systems with their potential to be used to treat oral lesions. The novelty of this manuscript lies with the specific polymers used and the ability of these polymer products (rather than drugs) to affect a number of diverse biological processes that appear to improve wound healing, although, it is unclear what advantages this material has over some of the other previously reported alternatives. The significance is therefore incremental overall but the study is well-conducted and does add weight and increasing credibility to this area of research.

C. Methodology/validity/robustness

The methodology used is appropriate in most instances. The quality and robustness of the dataset very good in many places, in particular the polymer chemistry, where there are few queries. However, I have significant concerns with some of the biological aspects of the manuscript which are outlined below. In addition, it seems that only one-way ANOVA was used for all statistical tests when n=3 was used. I have doubts if this would be normally distributed especially for the animal experiments?

C. Clarity, context, references

The manuscript could do with some language/grammar improvements/editing as there are many minor errors within the manuscript, although, overall, this does not detract from the information being described. There are many missing references regarding the use of mucoadhesive drug delivery systems to the oral mucosa for oral lesions that need to be included. This area of study has developed much in recent years but these studies have largely been ignored and this is a significant flaw in the manuscript. Some of the text within the figures is very small and difficult to read, especially the different combinations/ratios of polymers – this needs to be clearer as this information is the essence of the manuscript.

D. Suggested improvements/reviewer comments

1. The introduction needs radically revising, the information is not up-to-date with the literature and there are several important studies not mentioned that have a direct influence on this study. Indeed, the introduction does not mention any previously reported mucoadhesive patches for the oral cavity, some of which have strong evidence of efficacy.
2. Line 44 – there is very limited evidence to suggest that oral bacteria cause oral cancer due to infection of an oral ulcer. This is misleading and the statement should be retracted/alterd.
3. Lines 49-57. These sentences are mis-leading and does not reflect the current literature on oral patches where many polymer-based patches have been developed to adhere to the oral mucosa

for several hours with excellent drug delivery capabilities and high treatment efficacy (Colley et al Biomaterials 2018, 178:134-146; Teno et al (2022) J. Funct. Biomater. 13, 170; Paris et al. Acta Biomaterialia 2021, 128, 222-35; Liu et al, Small, 2022, 18, 2201561, Zhang et al, 2022 Materials & Design 223, 111131, Teno et al 2022 Functional Biomaterials 13:170, Alves 2020, Pharmaceutics 12:657) to name but only a few and these studies are more extensively reviewed in Edmans et al Pharmaceutics. 2020, 12(6):504; and Zhou et al, Biomolecules 2022, 12, 1254 amongst others. The text in the manuscript does not reflect the current situation regards mucoadhesive polymer patches and should be changed.

4. In the results for the patch fabrication: The hydrogen bonding interaction in scheme 1(iii) seems unlikely and does not appear to match the structure of the molecular dynamics simulation in Figure 1, which appears to show sodium ions acting as a bridge between multiple carboxylate oxygen atoms, although this is hard to make out.

5. The molecular dynamics simulations in figure 1 appear to be at different scales. Why were these not performed using the same simulation volume? The potential energy values are quoted per mole. It is confusing to understand what is meant by 1 mole of a binary mixture. Does the blend only have a more negative enthalpy because the value includes a greater number of monomer units or are the individual hydrogen bonding interactions actually stronger? This needs to be presented more clearly.

6. Could di-sulphide bonding contribute to the patch adhesive properties along with hydrogen/electrostatic bonding? – this seems a possibility. Can the authors comments on this?

7. In figure 3, the lap-shear tests are performed on skin tissue (human or animal?) with artificial saliva (type? Containing mucins?). Skin is highly keratinised and therefore has significantly different properties to the oral mucosa. Therefore, these experiments should be repeated but this time using oral mucosa and not skin – i.e., porcine mucosa which is similar to that of humans and accessible via abattoirs.

8. Line 254 polyLA content and adhesion – can this assumption be tested to validate your hypothesis?

9. I do not see the methods used for the durable wet tissue adhesion test (supplementary fig 16) was this performed with continuous agitation for 1-24 hours? Why was pig skin and not pig oral mucosa used?

10. Line 269 – “polyLA-Na/PolyLA-based adhesive patches all exhibit negligible swelling for the first 4 h...” the data in fig 4a shows very rapid swelling for around half the formulations with large increases in swelling ratio so this statement does not represent the data – please amend.

11. Release of PolyLA monomers following submersion in saliva – the data are provided as increases in absorbance in US spectra and % release rate Over 50 min. Can the actual amount of LA-Na monomer released be calculated and presented as this data would be very informative?

12. ROS inhibition assessment. PolyLA-Na monomer alone should be used as a positive control. Can prior disruption of the PolyLA-NA monomer be performed before addition to DPPH to further show that ROS inhibition is specifically due to this chemical species?

13. Fig 4f&g – what is the difference between images f and g (there appears to be no information in the methods concerning this experiment?). Why are the patches smaller in g than f? Also, this experiment contains no patch control. A positive control of PolyLA-Na monomer added in liquid form to an absorbent disc (used commonly in disc diffusion assays) would provide good evidence that the antibacterial effect is due to the released monomer and should be performed.

14. Gram-positive and Gram-negative bacteria have completely different cell walls, although it seems from fig h & j that inhibition of growth is instantaneous- can the authors explain why this is so – do the cells undergo instantaneous cell lysis? What is the mechanism of action? In addition, fig h/j and I/k do not match – in h/j growth is completely inhibited (0-5 min) while in i/k there is only 60% inhibition at 2 hours – why is this?

15. Line 329 – Gram-positive.

16. The minimum inhibitory concentration for the patch eluent against E. coli and S. aureus should be determined as this provides a better measure of antimicrobial activity?

17. The authors should state why chitosan was used as a control film (and also state the company where these patches were sourced in the materials and methods). It is difficult to determine the presence of the PolyLA-Na/PolyLA2-1 patch in the rat oral cavity at 6-24 h in fig 5d although this might be due to image quality. This needs to be clarified.

18. In vitro toxicity on fibroblast monolayers – was the same eluent concentration (1 mg/ml) used in these experiments as were conducted for the antimicrobial experiments? Did the authors test toxicity toward oral keratinocytes? In addition, biocompatibility should be investigated using cells

in direct contact with the material rather than only following exposure to the supernatant. For this tissue-engineered oral mucosal models should be used. These are commercially available and are much more representative of the oral mucosa than cells cultured as monolayers and toxicity experiments performed to OECD standards. The biopsy histological data for rat mucosal tissue would back up this data to confirm low toxicity.

19. Line 396 – “PolyLA-Na/PolyLA group is smoother” ...this is a subjective term that is not measured parameter, suggest changing this term.

20. How was the wound size and rate of closure calculated as this would depend on the angle the image was taken? Wounds treated with PolyLA-patch and imaged at 2 d displayed variable healing rate but this variability had dramatically declined at 4 d – can the authors explain this loss of experimental variability? Did the authors perform a power calculation to determine population size for rodent numbers in experiments (this is $n=3$ in the manuscript).

21. Fig 6 d/e/f the PolyLA-patch shows a remarkable effect on acid induced rodent ulcers which is very encouraging. The authors attempt to link this wound healing property to inhibition of leukocyte recruitment in patch-treated rats but this analysis is not robust. The immunostaining particularly for iNOS/MPO looks very non-specific and the images are of poor quality. There is no information on how this analysis was done – how many fields of view and sections analysed per rodent? Was the analysis done blind? The statistics state that ANOVA was used but how is the data normally distributed when $n=3$?

22. There is some good microbiology work conducted that shows differences between PolyLA-patch and chitosan-patch treatments. However, why would the microbiota in the control group who have had treatment with acid (a treatment designed to kill cells, including bacteria, thus creating an ulcer) be the same as in the normal (untreated) group. This does not seem intuitive? Can the authors comments on this.

23. In figure 8b, what is the yellow covering the wound in the control on day 3? This looks like a wound covering but there should not be any in this control pig?

24. How did the authors take into account the angle of the wound images in order to calculate the healing ration. For example, images at day 5 for control, PolyLA patch and Chitosan are taken at different angles which appear to show that the control has a larger wound size whereas the treated images are taken at a shallower angle, reducing the wound area?

25. The H&E and Masson-stained images look encouraging and strongly point towards quicker re-epithelialisation. However, the analysis of other markers such as Sirius red, CD31 and iNOS, IL-6, TNF is not robust and not well-defined in the methods section (e.g., much non-specific staining and the images are of poor quality. There is no information on how this analysis was done – how many fields of view and sections analysed per rodent? Was the analysis done blind? The statistics state that ANOVA was used but how is the data normally distributed when $n=3$?). Moreover, the potential effect on cytokine expression and angiogenesis is highlighted as a novel finding but these are based on observational findings. Further experiments are required here (e.g., tubule formation assays, in vitro endothelial cell, migration and proliferation assays, CAM assay, TNF/IL6/collagen gene expression assay on various cell types such as in vitro cultured macrophages, fibroblasts with and without Poly-LA eluant) are required here to confirm these initial preliminary observations.

26. The discussion is just a reiteration of the results, it does not place the study in context with other contemporary studies.

Responses to Reviewers' Comments

We thank the reviewers for their efforts. We feel we've significantly improved the content of the manuscript based on the reviewers' comments.

Reviewer #1:

This manuscript develops a hydrogel-based tissue adhesive for prolonged, sustainable oral ulcer treatment. Adopting hydrogel-based tissue adhesive to treat oral ulcer is novel to me. The overall material characterizations, in vitro demonstrations, and animal studies are comprehensive and effective. However, the key concern raised by the reviewer is that the intellectual merit of this work is limited. It is not clear why PolyLA-Na/PolyLA tissue adhesive outperforms many other tissue adhesives intensively studied in recent decades. The reviewer lists other comments in detail below.

Response: We are truly grateful to the reviewer's comments on our work. In recent years, tissue adhesives have developed rapidly and have been widely used in various fields. However, most of the reported tissue adhesives can only adhere to dry tissue, and the adhesion strength of these adhesives will significantly decrease or even disappear due to the weak interactions and bulk instability in wet environment.^[1-3] Although some strategies have been proposed to promote the adhesion of the adhesive to the wet tissue interface, such as patterning the surface of the adhesive^[4,5], introducing hydrophobic structure into the adhesive^[6,7], topological adhesion^[8,9], dry polymer film adhesion^[10,11] and solvent exchange^[12], these adhesives have some defects, such as complex preparation process, poor biocompatibility, no immediate adhesion and non-degradation in vivo. Inspired by marine mussels, catechol-based wet tissue adhesives have also been extensively studied, but due to the structural instability of catechol, its properties are affected by temperature, pH value, oxygen content and so on. This greatly limits the in vivo use of these adhesives.^[13-15] In addition, these adhesives often do not have biological functions, they can only play the role of physical seal when used in vivo, and cannot effectively promote tissue

repair and regeneration. In order to give biological function to these adhesives, drugs, growth factors or cells need to be loaded in the adhesives, which may bring potential drug toxicity to organisms and significantly increase the cost of materials. In order to overcome these defects of tissue adhesives and promote the translation of tissue adhesives to clinical application, the adhesion strength, convenient preparation, safe source of raw materials and biocompatibility of tissue adhesive need to be considered. This inspires us to continue to create tissue adhesives from bioactive molecules occurring in the body. α -Lipoic acid (LA) is a coenzyme found in mitochondria, and participates in mitochondrial activity and synergistic energy metabolism. Moreover, LA shows an excellent antioxidant, anti-inflammation, antibacterial and angiogenesis regeneration activity.^[16-18] Because of its unique biological activity, LA has been widely applied in nanomedicine for the treatment of diabetes, Alzheimer's disease, and cancer.^[19-21] All these show that LA has a good biosafety and bioactivity. Based on the ring-opening self-polymerization mechanism initiated by dynamic disulfide bond exchange and abundant adhesive carboxyl groups of LA, it has been developed as supramolecular polymer adhesives. However, the as-polymerized poly(α -Lipoic acid) (PolyLA) is metastable due to the inverse ring closing depolymerization initiated by terminal reactive radicals. Although stable PolyLA-based adhesives can be obtained by introducing multiple double-bond monomers, multivalent metal ions or ionic liquids into the system, the lack of effective purification methods and poor biocompatibility restrain the biomedical applications of these adhesives^[22-24]. And the stabilized PolyLA and the natural hydrophobicity of LA restrict the release of bioactive TA monomer from these adhesives when used in vivo, sacrificing its biofunctions. Unlike LA, sodium α -lipoate (LA-Na) is an amphiphilic small molecule with high solubility in water and PolyLA-Na is stable at room temperature based on its highly ordered crystal structure^[25]. However, PolyLA-Na is highly sensitive to water, and it will quickly dissociate into small molecule LA-Na after contact with water. Although LA-based small molecules can be released, the rapid dissociation rate still limits its application in vivo (less than 20 min) and it is often used to prepare degradable plastics. Therefore, in this work, we engineered a self-stabilized and

water-responsive deliverable binary synergistic adhesive patch made from coenzyme salt polymer for the first time. The synergism between PolyLA and PolyLA-Na not only inhibited the reversible depolymerization of PolyLA at room temperature, but also reduced the dissociation rate of PolyLA-Na in wet environment and prolonged the release time of bioactive LA-Na molecules whose release could be triggered by water (from less than 20 minutes to 12 hours). Meanwhile, this PolyLA-Na/PolyLA binary elastic and extensible patch exhibited robust wet tissue adhesion due to the adhesive action of PolyLA, while the hydrophilic PolyLA-Na network can remove the hydration layer (PolyLA-Na/PolyLA adhesive patch could firmly adhere to the wet surface of oral mucosa *in vivo* for about 24 h, while commercial chitosan patch adhered to the surface of oral mucosa for only less than 2 h). We used PolyLA-Na/PolyLA as oral ulcer repair patch. The results of animal experiments showed that PolyLA-Na/PolyLA patch could stably and durably adhere to the surface of oral mucosa to resist the impact of saliva or swallowing, and could significantly accelerate the repair of oral ulcer defect by promoting epithelial and angiogenesis compared with commercial chitosan oral repaired patch. Although other tissue adhesives have been reported to be used to repair oral mucosal lesions, these adhesives also have some shortcomings such as complex preparation process, exogenous drug loading, photothermal initiation and unreacted toxic small molecules.^[26-28] Therefore, in this work, we used biosafe and bioactive LA-based monomers to prepare a self-stable PolyLA-based tissue adhesive patch in a very simple way for the first time, and applied it *in vivo* without loading any therapeutic drugs. This will not only provide a new idea for the preparation of PolyLA-based adhesive, but also open up a new avenue for constructing biofunctional tissue adhesives from naturally occurring bioactive molecules.

1. The authors systematically compare PolyLA-Na/PolyLA, PolyLA, and PolyLA-Na tissue adhesives. However, there is little discussion on how the tissue adhesive compared with many other tissue adhesives recently studied (H. Yuk et al., *Nature*, 575, 2019; H. Yuk et al., *Nature Biomedical Engineering*, 5, 2021; J. Li et al., *Science*,

357, 2017). The reported adhesion strength in these literature is about 100 kPa, much larger than this work.

Response: We thank the reviewer's comment. We believe that the design of tissue adhesive should not be limited to adhesion strength, and its functionalization bioactivity should also be considered. Moreover, the required adhesion strength of tissue adhesive is different in different application scenarios, as long as it meets the application requirements. In the revised manuscript, we re-supplemented the adhesion strength of PolyLA-Na/PolyLA-based adhesive patch to oral mucosal tissue. The adhesive patch can achieve immediate high-strength adhesion to moist oral mucosal tissue, and the immediate adhesion strength can reach 60 kPa, which is sufficient for the application of the patch in oral cavity. Moreover, as shown in Figure 5a-d in the original manuscript, the PolyLA-Na/PolyLA binary synergistic adhesive patch can firmly adhere to all parts of the rat oral cavity, and shows a more lasting adhesion behavior compared with the commercial chitosan oral repaired patch, which proves that the adhesion strength of the PolyLA-Na/PolyLA patch meets the needs of oral ulcer patch. In addition, our binary synergistic adhesive patch also shows a variety of biological functions and significantly accelerates the repair of oral ulcers. Those studies mentioned by the reviewer showed higher wet tissue adhesion strength, but these adhesives may face biosafety issues due to the shedding of -NHS activated groups and the non-biodegradability of polyacrylic acid or polyacrylamide. Moreover, those adhesives lacked biological function, which is essential for tissue repair and regeneration. In order to demonstrate the advantages in oral repair, we compared our PolyLA-Na/PolyLA binary synergistic adhesive patch with other reported oral adhesive patches, and supplemented it in the revised manuscript. The supplementary content has been marked red in the manuscript. As shown in **Table R1**, although a large number of oral adhesive patches have been developed in recent years, these adhesive patches often need to be loaded with exogenous drugs, factors or proteins to promote the repair of oral injuries, and they have no therapeutic effect in themselves. Moreover, those patches show a short residence time in vivo, and the adhesion process needs light irradiation to induce gelation, which will also greatly limit the

application efficiency of these patches in vivo.

Table R1. Comparison of PolyLA-Na/PolyLA-based patch with previous work (/ indicates parameters that were not mentioned in the reported work).

Materials	Load component	Adhesion time in vivo	Adhesion strength/mode
PolyLA-Na/PolyLA based patch	Without any exogenous components	> 12 h	60 kPa/slight pressing for 5 s
3D printing PAA-DOPA-CMC patch ^[29]	Oxaliplatin/Mycophenolate	4 h	1.29 ± 0.30 N/5 N pressing
Chitosan-HA patch ^[30]	Ovalbumin	/	/
PVP-PEP bilayer electrospun patch ^[31]	Clobetasol-17-propionate	4 h	/
PVP-Eudragit electrospun patch ^[32]	anti-TNF α	3 h	/
PCL-PLA-PEO-multilayer electrospun patch ^[33]	Ciprofloxacin hydrochloride	7 h	/
PCL-Gelatin-PGA electrospun patch ^[34]	Astaxanthin	2 h	/
HA-CNB Gel ^[26]	/	24 h	70 kPa/UV illumination
Gelatin-PDA-nanoclay hydrogel ^[35]	Dexamethasone	/	63 kPa/pressing for 30 s
PVA-DOPA film ^[27]	Dexamethasone	> 4 h	38.72 ± 10.94 kPa
GelMA-TA-nanoclay hydrogel ^[36]	/	10 h	20 kPa
PACG-MeGG hydrogel ^[28]	/	4 h	3 kPa
PAA-chitosan dry gel ^[37]	5-aminolevulinic acid	/	40 kPa

2. The intellectual merit of the material system is limited. The key design of this work introduces hydrogen bonds between PolyLA-Na and PolyLA to enable a prolonged degradation, resulting in sustainable drug delivery. This design is obvious to the reviewer.

Response: We thank the reviewer's comment. We think that the reviewer may have misunderstood our work, and as we described in the Introduction section of the manuscript, pure PolyLA is unstable at room temperature and will depolymerize to opaque and non-adhesive oligomer due to its terminal sulfur free radicals (Figure 1a

in the original manuscript). Although it has been reported that the introduction of multiple double-bond monomers, multivalent metal ions or ionic liquids into the PolyLA system can stabilize PolyLA, due to the lack of effective purification methods, the release of toxic small molecules and poor biocompatibility make these adhesives unable to be applied in vivo^[22-24]. In addition, the stabilized PolyLA and the natural hydrophobicity of LA restrict the release of bioactive TA monomer from these adhesives when used in vivo, sacrificing its biofunctions. But for LA-Na, it shows amphiphilic characteristics, which can be dissolved in water, and with the volatilization of water, LA-Na is spontaneously opened and self-assembled into a highly ordered PolyLA-Na network. Moreover, PolyLA-Na network exhibits a water sensitive dissociated behavior. When PolyLA-Na comes into contact with water, it can be quickly dissociated into small lipoic acid salts^[25,38]. Although active small molecules LA-Na can be effectively released, the excessively rapid dissociation rate (less than 20 min) (Supplementary Fig. 6 in the original manuscript), non-adhesive and rigid nature of PolyLA-Na greatly limit its direct application in the biomedical field. Therefore, PolyLA-Na is only used to prepare biodegradable plastics, but does not show its value in the biomedical field. In this work, by simply blending PolyLA and PolyLA-Na, the hydrogen bond between them cannot only reduce the potential energy of the system, make PolyLA stable and adhesive at room temperature, but also reduce the water sensitivity of PolyLA-Na and prolong the release time of LA-Na active small molecules. This is the first time that we propose to employ the salt structure of LA to stabilize PolyLA without introducing any other exogenous small molecules. Thus, this design has the effect of "killing two birds with one stone", rather than simply prolonging the dissociation rate of PolyLA-Na in water. This provides a new idea to stabilize coenzyme polymer by introducing its own salt, which could maintain the bioactivity of the coenzyme without resorting to exogenous substances.

3. In Line 99 on Page 5, "... the amorphous PolyLA serves to reduce the crystallinity of PolyLA-Na, thus increasing the elasticity and extensivity of the PolyLA-Na film." To clarify this physical picture, the authors are recommended to provide stress-strain

curves of materials in Fig. 2. The y-axis in Fig. 2g-2i should be strength instead of stress. The y-axis in Fig. 2j-2l should be ultimate strain instead of strain.

Response: We thank the reviewer's comment. In fact the stress-strain curves of PolyLA-Na/PolyLA binary synergistic adhesive patches of different compositions were placed in Supplementary Fig. 13 in the original manuscript. The results of stress-strain curves also proved that the PolyLA-Na/PolyLA binary synergistic system transforms from a typical crystalline polymer to an amorphous polymer with the increasing the content of PolyLA. According to the suggestions of the reviewers, we have modified the y-axis of Fig.2g-2i and Fig. 2j-2l in the revised manuscript.

4. In Line 106 on Page 5, "...the adhesive PolyLA hydrophobic network is still maintained...", which is unclear. What is the crosslinker for the PolyLA hydrophobic network?

Response: The maintenance of PolyLA network framework in aqueous environment mainly depends on the multiple hydrogen bond cross-linking between carboxyl groups in PolyLA polymer chain. Fig. 4c in the original manuscript proved that although LA-Na molecules are released after immersion in water of PolyLA-Na/PolyLA binary synergistic adhesive patch, there are still a large number of carboxyl groups on the PolyLA network, and the hydrogen bond interaction between carboxyl groups is conducive to the stability of PolyLA in water. In addition, the hydrophobic nature of PolyLA itself also help to maintain its stability in the water environment. We have added the description of this part in the manuscript, and the supplementary content has been marked red.

5. The authors discussed the crystalline-to-amorphous transition. To quantify the crystallinity, the authors are recommended to perform differential scanning calorimetry to measure the crystallinity in the series of samples directly.

Response: We thank the reviewer's comment. According to the suggestion of the reviewer, we carried out DSC tests on different components of PolyLA-Na/PolyLA patches, but due to the reversible depolymerization of PolyLA under heating, the

patch was depolymerized while melting, so there are multiple fluctuating peaks instead of a single melting peak in the DSC curve (**Fig. R1**). Thus we could not use DSC to calculate the crystallinity of the patch. In order to prove the transformation from crystallization to amorphous structure of PolyLA-Na/PolyLA patches with different composition, XRD tests were carried out on the PolyLA-Na/PolyLA based patches, and the crystallinity of the patch with different composition was calculated. From **Fig. R2** we can see that the sharp crystallization peak at a small angle gradually decreases and disappears with the increase of PolyLA content. The crystallinity of PolyLA-Na/PolyLA patches with different composition was calculated by using the ratio of the area of the crystallization peak to the total area in the XRD curve. As shown in **Fig. R3**, for pure PolyLA-Na, it is a highly crystalline structure with a crystallinity of 91%. After introducing PolyLA into PolyLA-Na system, the crystallinity of the patch decreased obviously, and the crystallinity of PolyLA-Na/PolyLA-1.5-0.5 was 13.5%. Further increasing the content of PolyLA, the crystallinity of the patch further decreased, and the crystallinity of PolyLA-Na/PolyLA-2-1 and PolyLA-Na/PolyLA-3-1 was only 1.4% and 0.7%, indicating that the PolyLA-Na/PolyLA-based patch gradually changed from crystalline structure to amorphous structure with the increase of PolyLA content.

Fig. R1 DSC curves of PlyLA-Na/PolyLA patch with different compositions.

Fig. R2 XRD curves of PlyLA-Na/PolyLA patch with different compositions.

Fig. R3 Crystallinity of PolyLA-Na/PolyLA patch with different compositions.

6. Some fonts in the main figures are too small to be visualized. The authors are suggested to revise the fonts in Fig 2, 3, 4, 7. The data presented in Fig. 4 is too crowded. The authors can keep the most critical data in Fig. 4 while leaving detailed raw data in supplementary materials. The fonts in Fig. 7f and 7g are too small.

Response: We thank the reviewer's comment. We have enlarged the font in the figures in the original manuscript, and only the data of PolyLA-Na/PolyLA-1.5-1, PolyLA-Na/PolyLA-2-1 and PolyLA-Na/PolyLA-3-1 are retained in Fig. 4a in the revised manuscript, and the other data were transferred to the supplementary file (Supplementary Fig. 18, 20 and 21). For Fig. 7, we have adjusted the sharpness of the picture to provide the clearest and intuitive data for reviewers.

Reviewer #2:

This study, the authors presented the strategy for preparing an oral mucosa adhesive patch consisting of PolyLA and PolyLA-Na networks. The binary synergistic PolyLANa/PolyLA patch showed durable adhesion to oral mucosal wound and water-responsive sustainable release of bioactive small molecules. In treating oral mucosa defect, the adhesive patch showed considerably improved therapeutic efficacy. This paper is well-organized with systematic experimental data and logical language description. I recommend it to be published in the journal of Nature Communications after the following issues be addressed:

Response: We are truly grateful to the reviewer's comments on our work, especially for the constructive suggestions.

1. Measurement of adhesion strength of PolyLA-Na/PolyLA adhesive patches to skin tissue, does the sample thickness affect the adhesion strength/adhesion force? Additionally, the source of the skin tissue should be specified.

Response: We thank the reviewer's comment. According to the opinions of other reviewers, in order to more truly simulate the environment of PolyLA-Na/PolyLA binary synergistic adhesion patch in oral cavity, we used pig oral mucosa tissue instead of pig skin to test the adhesion strength. The results showed that the adhesion strength of PolyLA-Na/PolyLA binary synergistic adhesion patches to mucosal tissue was significantly stronger than that of pig skin, and the highest adhesion strength to oral mucosal can reach 60 kPa, while the highest adhesion strength to pig skin is only 15 kPa in the Fig. 3h of original manuscript (**Fig. R4a**), which may be caused by the different degree of keratinization and the content of protein or collagen on the surface of mucous membrane and skin. This means that our PolyLA-Na/PolyLA binary synergistic adhesion patch is suitable for the repair of oral mucosal injury. In order to further verify the effect of material thickness on its adhesion to tissue, we prepared PolyLA-Na/PolyLA-2-1 patches with a thickness of 0.5, 0.8 and 1 mm, respectively, and tested the instant adhesion strength of these materials to oral mucosa. As shown in Fig. R4b, there was no significant difference in the instant adhesion strength of the three thickness patches to the oral mucosa, indicating that the thickness of the patch had little effect on its adhesion properties. In order to ensure the comfort used in the oral cavity, we choose the patch with the thickness of 0.5 mm for animal experiment. The oral mucosa of pigs was obtained from the local slaughterhouse, and this information has been added to the revised manuscript.

Fig. R4 (a) Adhesion strength of PolyLA-Na/PolyLA adhesive patches with different composition to pig oral mucosa tissue (n=4); (b) Adhesion strength of PolyLA-Na/PolyLA-2-1 patch with different thickness to pig oral mucosa tissue (n=4).

2. The method section shows that the animals were treated with the patch every two days after the first dressing, but the authors only displayed that the patches could maintain stable adhesion with skin tissue after being soaked in artificial saliva (Supplementary Fig. 16) and with oral mucosa of rat (Fig. 5d) for 24 hours, how about the patch state for 48 hours. How long can the patch adhere stably to oral mucosa?

Response: We thank the reviewer's comment. As indicated by Fig.5d in the original manuscript, because there are a large number of active enzymes in the oral cavity, the PolyLA-Na/PolyLA patch will dissociate gradually after adhering to the rat oral mucosa surface, and basically dissociate completely after 24 h, so the patch can only exist on the rat oral mucosa surface for 24 h. However, it can be seen from the result (Fig. 5d in the original manuscript) that the in vivo stability and adhesion persistence of PolyLA-Na/PolyLA patch is much higher than that of commercial chitosan oral patch, and chitosan patch will be completely dissociated within 2 h in oral cavity of rat. In in vitro experiment, due to the inability to truly simulate the oral environment and the lack of active substances, the patch will not dissociate quickly after soaking in the artificial saliva, and it can maintain adhesion to the wet tissue surface after soaking in artificial saliva for 24 h even in agitated environment. Further extending the soaking time in vitro, the patch can still adhere to the tissue surface, but the tissue will deteriorate due to the hot and humid environment, which is not conducive to

further observation. In this work, 48 h was chosen as the dressing change cycle, which was to reduce the effect of anesthesia on the mental and life state of rats. If the dressing change time is shortened, the anesthetic effect may threaten the life of rats.

3. The adhesive strength of the PolyLA-Na/PolyLA adhesive patch to the artificial saliva-treated skin tissue was tested before swelling of the patch. The adhesion strength of the patch after swelling is suggested to test and discuss, as the oral environment is aquiferous.

Response: We are very grateful to the reviewers' suggestion. We noted that the PolyLA-Na/PolyLA patch cannot show adhesion ability after swelling, because after the patch is soaked in liquid, the PolyLA-Na skeleton dissociates and the carboxyl group on the PolyLA skeleton is hydrated, so the patch loses the function of drainage and adhesion. However, once the PolyLA/PolyLA-Na patch adheres to the tissue, even if it swells in the wet environment, it still firmly adheres to the tissue surface. The reason is that the patch and tissue have formed a strong hydrogen bond interaction. Therefore, even in the wet oral environment, the patch still can maintain a lasting and stable adhesion on the surface of the oral mucosa. In order to verify the durable adhesion stability of PolyLA-Na/PolyLA patch to tissue when used in high humidity and dynamic oral environment, we completely soaked the PolyLA-Na/PolyLA-2-1 patch adhered porcine oral mucosa in artificial saliva, and stirred at 37 °C at a speed of 120 r/min for 24 h. We found that the patch still firmly adhered to the oral mucosal tissue even in high wet and dynamic environment for a long time and could lift the weight of the mucosal tissue itself (**Fig. R5**). This means that PolyLA-Na/PolyLA patch has a stable adhesion ability to oral mucosal tissue in oral environment.

Fig. R5 Adhesion stability of PolyLA-Na/PolyLA patch to oral mucosa by imitating

oral environment.

4. Does the monomer release affect the adhesive strength of the PolyLA- Na/PolyLA adhesive patch. The adhesive strength of the patch after soaking in artificial saliva for different times (e.g. 12, 24, 36, 48 h) is suggested to measured.

Response: We thank the reviewer's comment. Considering the deterioration of tissue itself in artificial saliva environment at 37 °C for a long time and the stability of patch in vivo, we tested the adhesion strength of the PolyLA-Na/PolyLA-2-1 patch to pig oral mucosa after soaking in artificial saliva for 3, 6, 12 and 24 h (**Fig. R6**). The results showed that the adhesion strength of PolyLA-Na/PolyLA-2-1 patch decreased significantly after soaking in artificial saliva. When the soaking time reached 24 h, the adhesion strength of the patch to oral mucosa was 4.6 kPa. The decrease of adhesion strength of PolyLA-Na/PolyLA patch after soaking in artificial saliva may be due to the continuous dissociation of PolyLA-Na network and the hydration of adhesive carboxyl groups in the patch, but even if the adhesion strength decreases after soaking in artificial saliva environment, the patch can still adhere to the surface of oral mucosa tissue (Fig. R5).

Fig. R6 Adhesion strength of PolyLA-Na/PolyLA-2-1 to oral mucosa tissue after soaking in artificial saliva for different times.

5. In treatment of mucosa defect model in mini-pig, the sizes of the PolyLANa/PolyLA-2-1 patch and chitosan film should be specified?

Response: We thank the reviewer's comment. In the treatment of oral mucosa defect model in mini-pig, both PolyLA-Na/PolyLA-2-1 and chitosan patches were disc patches with a diameter of 1.2 cm. The relevant description has been added to the revised manuscript and has been marked red.

Reviewer #3 :

A. Key results

This interesting study by Cui et al describes the production of a polymer mucoadhesive patch based on PolyLA-Na and PolyLA that provides the basis for the release of poly alpha-lipoic acid, which the authors suggest, has a range of diverse biological activity including antimicrobial, angiogenic whilst also regulating the immune response and increasing wound healing. These mucoadhesive patches were tested on experimentally induced oral ulcers/wounds in mice and pigs to show efficacy in vivo.

Response: We are truly grateful to the reviewer's comments on our work, especially for the constructive suggestions.

B. Significance

There are now many reports going back several years on of the use of fabricated polymer-based mucoadhesive patches based on gels, electrospun fibers etc. for the treatment of oral ulcers or other oral lesions. These reports differ on the polymers used, type of fabrication process and therapeutic drug/protein delivered (Colley et al Biomaterials 2018, 178:134-146; Teno et al (2022) J. Funct. Biomater. 13, 170; Paris et al. Acta Biomaterialia 2021, 128, 222-35; Liu et al, Small, 2022, 18, 2201561, Zhang et al, 2022 Materials & Design 223, 111131, Teno et al 2022 Functional Biomaterials 13:170, Alves 2020, Pharmaceutics 12:657). There are reports of these progressing to the clinic, completing phase 2 clinical trials (Ibrahim et al. BMC Oral Health (2023) 23:99; Brennan et al 2021 DOI: 10.1111/jop.13270). Moreover, some reports have showed that the polymers themselves may also have biological effects (Edmans 2022, doi.org/10.1016/j.jconrel.2022.08.016). This manuscript adds to the

increasing list of mucoadhesive polymers and systems with their potential to be used to treat oral lesions. The novelty of this manuscript lies with the specific polymers used and the ability of these polymer products (rather than drugs) to affect a number of diverse biological processes that appear to improve wound healing, although, it is unclear what advantages this material has over some of the other previously reported alternatives. The significance is therefore incremental overall but the study is well-conducted and does add weight and increasing credibility to this area of research.

Response: We are truly grateful to the reviewer's comments. As mentioned by the reviewer, tissue adhesives have developed rapidly in recent years, in which oral adhesive patches have been widely studied, and many studies on oral adhesive patches have also been reported. These patches are often fixed on the surface of oral mucosa by physical adsorption, and promoted oral mucosal wound healing by loading exogenous drugs, cell or anti-inflammatory factor. However, only physical adsorption is often unable to maintain the long-term adhesion of the patch to the wet oral mucosal surface (most of the reported oral mucoadhesion patches adhere to oral mucosa surface in vivo for about 4 h, and a small number of reported mucoadhesive patch can stay on the oral mucosa for 7 h and 10 h), and the loaded exogenous substance may have potential drug toxicity and increase the cost of materials. Although some of these studies have confirmed the effectiveness of patches through volunteers, or clinical trials have been gradually carried out, it is also very important to develop durably adhesive and intrinsically bioactive patches to facilitate the repair of oral mucosal ulcer. Here, we compared the performance of our PolyLA-Na/PolyLA patch with the reported mucohesive patches (Table R1). Therefore, in this work, we propose a new design strategy for creating oral adhesive patch in a very simple way without loading any exogenous drugs or biological factors, from bioactive molecules occurring in the body. In addition, the very important significance of our work is the discovery of a new method for self-stabilization of PolyLA, which has not been reported before. By simply mixing PolyLA and PolyLA-Na, inhibiting the ring-opening depolymerization of PolyLA, reducing the dissociation rate of PolyLA-Na in aqueous environment, realizing the transformation of polymer from

crystallization to amorphous structure and maintaining biological functions of coenzyme. This work provides a new idea to stabilize coenzyme polymer by introducing its own salt, which could maintain the bioactivity of the coenzyme without resorting to exogenous substances.

Table R1. Comparison of PolyLA-Na/PolyLA-based patch with previous works (/ indicates parameters that were not mentioned in the reported work).

Materials	Load component	Adhesion time in vivo	Adhesion strength/mode
PolyLA-Na/PolyLA based patch	Without any exogenous components	> 12 h	60 kPa/slight pressing for 5 s
3D printing PAA-DOPA-CMC patch ^[29]	Oxaliplatin/Mycophenolate	4 h	1.29 ± 0.30 N/5 N pressing
Chitosan-HA patch ^[30]	Ovalbumin	/	/
PVP-PEP bilayer electrospun patch ^[31]	Clobetasol-17-propionate	4 h	/
PVP-Eudragit electrospun patch ^[32]	anti-TNF α	3 h	/
PCL-PLA-PEO-multilayer electrospun patch ^[33]	Ciprofloxacin hydrochloride	7 h	/
PCL-Gelatin-PGA electrospun patch ^[34]	Astaxanthin	2 h	/
HA-CNB Gel ^[26]	/	24 h	70 kPa/UV illumination
Gelatin-PDA-nanoclay hydrogel ^[35]	Dexamethasone	/	63 kPa/pressing for 30 s
PVA-DOPA film ^[27]	Dexamethasone	> 4 h	38.72 ± 10.94 kPa
GelMA-TA-nanoclay hydrogel ^[36]	/	10 h	20 kPa
PACG-MeGG hydrogel ^[28]	/	4 h	3 kPa
PAA-chitosan dry gel ^[37]	5-aminolevulinic acid	/	40 kPa
Tacrolimus patch ^[53]	Tacrolimus	/	/

C. Methodology/validity/robustness

The methodology used is appropriate in most instances. The quality and robustness of the dataset very good in many places, in particular the polymer chemistry, where there are few queries. However, I have significant concerns with some of the biological aspects of the manuscript which are outlined below. In addition, it seems that only

one-way ANOVA was used for all statistical tests when $n=3$ was used. I have doubts if this would be normally distributed especially for the animal experiments?

Response: We are truly grateful to the reviewer's comments. We are very sorry that we left out the statistical analysis method about animal experiments in the statistical analysis part. With regard to the data related to animal experiments, we choose the student's t-test to examine the differences between the two groups of data rather than the variance test. We are very sorry for the confusion caused to you by our mistakes, we have re-described and supplemented the statistical analysis in the revised manuscript, and marked the revised content red.

D. Clarity, context, references

The manuscript could do with some language/grammar improvements/editing as there are many minor errors within the manuscript, although, overall, this does not detract from the information being described. There are many missing references regarding the use of mucoadhesive drug delivery systems to the oral mucosa for oral lesions that need to be included. This area of study has developed much in recent years but these studies have largely been ignored and this is a significant flaw in the manuscript. Some of the text within the figures is very small and difficult to read, especially the different combinations/ratios of polymers – this needs to be clearer as this information is the essence of the manuscript.

Response: We are truly grateful to the reviewer's comments. We have carefully examined and revised the language, grammar and editing of the manuscript, and supplemented the related research work of mucoadhesive drug delivery systems reported in recent years; in addition, we modified some figures of original manuscript to make the text in the pictures appear more clearly. The revised and supplementary contents have been marked red in the revised manuscript.

D. Suggested improvements/reviewer comments

1. The introduction needs radically revising, the information is not up-to-date with the literature and there are several important studies not mentioned that have a direct

influence on this study. Indeed, the introduction does not mention any previously reported mucoadhesive patches for the oral cavity, some of which have strong evidence of efficacy.

Response: We are truly grateful to the reviewer's comments that will help improve the quality of our manuscript. We have supplemented and discussed the related studies on oral mucoadhesive patches reported in recent years in the Introduction part of the revised manuscript. The supplemented and revised contents have been marked red.

2.Line 44–there is very limited evidence to suggest that oral bacteria cause oral cancer due to infection of an oral ulcer. This is misleading and the statement should be retracted/altered.

Response: We thank the reviewer's comment. We modified this sentence in the revised manuscript and deleted the related description of oral cancer.

3.Lines 49-57. These sentences are mis-leading and does not reflect the current literature on oral patches where many polymer-based patches have been developed to adhere to the oral mucosa for several hours with excellent drug delivery capabilities and high treatment efficacy (Colley et al Biomaterials 2018, 178:134-146; Teno et al (2022) J. Funct. Biomater. 13, 170; Paris et al. Acta Biomaterialia 2021, 128, 222-35; Liu et al, Small, 2022, 18, 2201561, Zhang et al, 2022 Materials & Design 223, 111131, Teno et al 2022 Functional Biomaterials 13:170, Alves 2020, Pharmaceutics 12:657) to name but only a few and these studies are more extensively reviewed in Edmans et al Pharmaceutics. 2020, 12(6):504; and Zhou et al, Biomolecules 2022, 12, 1254 amongst others. The text in the manuscript does not reflect the current situation regards mucoadhesive polymer patches and should be changed.

Response: We thank the reviewer's comment. We have revised the part of Introduction, and added the cutting-edge progress of oral adhesive patch in the revised manuscript.

4.In the results for the patch fabrication: The hydrogen bonding interaction in scheme

1(iii) seems unlikely and does not appear to match the structure of the molecular dynamics simulation in Figure 1, which appears to show sodium ions acting as a bridge between multiple carboxylate oxygen atoms, although this is hard to make out.

Response: We thank the reviewer's comment. Because the -COO^- on the PolyLA-Na polymer chain has stronger electron withdrawal property than the -COOH on the PolyLA polymer chain, the hydrogen bond interaction between -COO^- and -COOH is stronger than that of the two -COOH s. In order to show the hydrogen bond interaction between -COO^- and -COOH more accurately, we reviewed the relevant studies^[39,40] and modified the hydrogen bond interaction in Scheme 1(iii) in the revised manuscript. In this work, sodium ion does not act as a bridge between multiple carboxyl oxygen atoms. It is the display problem in the results of molecular dynamics simulation that causes the visual illusion that sodium ion can act as a bridge between carboxyl oxygen atoms. We have adjusted the position of sodium ions in the result of molecular dynamics simulation in the revised manuscript.

5.The molecular dynamics simulations in figure 1 appear to be at different scales. Why were these not performed using the same simulation volume? The potential energy values are quoted per mole. It is confusing to understand what is meant by 1 mole of a binary mixture. Does the blend only have a more negative enthalpy because the value includes a greater number of monomer units or are the individual hydrogen bonding interactions actually stronger? This needs to be presented more clearly.

Response: We thank the reviewer's comment. In this simulation system, in order to compare the stability and potential energy of PolyLA system, PolyLA-Na system and their mixed system at room temperature clearly, we deliberately keep the same number of the same molecule in different cell to ensure that the basis of comparison is consistent. Moreover, the simulations of the above three systems are all carried out at 300 K and 1 atm, and the density of the three systems keeps equilibrium, so the simulation results are comparable. In addition, we have verified that the hydrogen bond interaction between -COO^- of LA-Na and -COOH of LA is stronger than that between two -COOH s of LA through molecular dynamics simulations in other work.

It can be seen from the **Fig R7** that the bonding energy of the hydrogen bond between LA and LA-Na is -44.3 kcal/mol, while the bonding energy of the hydrogen bond between LA dimer is -20.64 kcal/mol, indicating that the hydrogen bond between LA and LA-Na (-COOH \cdots O=C-O-) was stronger than the hydrogen bond between LA dimer (-COOH \cdots HOOC-) though the latter could form two hydrogen bonds. Therefore, the reason why the potential energy of the mixed system of PolyLA-Na and PolyLA is lower than that of the pure PolyLA-Na system and the PolyLA system is that there is a stronger hydrogen bond interaction between PolyLA-Na and PolyLA. Therefore, the lower potential energy of the mixed system is due to the stronger hydrogen bond interaction between PolyLA-Na and PolyLA rather than the number of molecules in the system.

Fig. R7 Molecular dynamics simulation of bond energy of hydrogen bond formed between LA and LA-Na as well as LA dimer.

6. Could di-sulphide bonding contribute to the patch adhesive properties along with hydrogen/electrostatic bonding?—this seems a possibility. Can the authors comment on this?

Response: We thank the reviewer's comment. At present, few studies have reported the use of disulfide bonds to achieve tissue adhesion. Although the dynamic exchange of disulfide bonds may form covalent bonds with sulfhydryl groups of tissue interface, this interaction has been rarely investigated and studied in tissue adhesives. In this work, we think that the adhesion based on disulfide bond can be ignored, because the PolyLA-Na/PolyLA binary synergistic adhesive patch we prepared is relatively stable. However, the adhesion mechanism proposed by the reviewer is worthy of our in-depth study in the future work.

7. In figure 3, the lap-shear tests are performed on skin tissue (human or animal?) with artificial saliva (type? Containing mucins?). Skin is highly keratinised and therefore has significantly different properties to the oral mucosa. Therefore, these experiments should be repeated but this time using oral mucosa and not skin—i.e., porcine mucosa which is similar to that of humans and accessible via abattoirs.

Response: We thank the reviewer's comment. The skin tissue used in the adhesion strength test in this work is pig skin tissue. And the artificial saliva we use in this work is in line with the international certification standard of experimental artificial saliva (ISO10271), which is a standard ionic saliva that does not contain mucin and other organic components.

We are very grateful to the reviewer for the suggestions on the determination of adhesion strength in the manuscript. We re-investigated the adhesion properties of PolyLA-Na/PolyLA binary synergistic adhesive patches by using pig oral mucosa tissue instead of pig skin tissue as suggested by the reviewers. To our surprise, the adhesion strength of PolyLA-Na/PolyLA binary synergistic adhesive patches to pig oral mucosa was significantly higher than that to pig skin, which may be due to the different degree of keratinization and the different content of protein or collagen on the surface of mucous and skin. From the **Fig. R8a**, we can see that the highest adhesion strength of PolyLA-Na/PolyLA patch to oral mucosa tissue can reach 60 kPa, but the highest adhesion strength to pig skin is only 15 kPa (Fig. 3h in original manuscript). From the process of lap shearing (Fig. R8b), it can also be seen that the PolyLA-Na/PolyLA patch adheres tightly to the oral mucosal tissue and shows the cohesion failure of the patch during the test. We have revised and supplemented the relevant content in the manuscript, and the revised content has been marked red.

Fig. R8 (a) Adhesion strength of PolyLA-Na/PolyLA adhesive patches with different composition to pig oral mucosa tissue after soaking in artificial saliva (n=4); (b) Photographs showing the lap-shear process of the PolyLA-Na/PolyLA-2-1 patch adhered to pig oral mucosa tissue after soaking in artificial saliva.

8.Line 254 polyLA content and adhesion—can this assumption be tested to validate your hypothesis?

Response: We thank the reviewer’s comment. The adhesion strength of our supplementary PolyLA-Na/PolyLA binary synergistic adhesive patches to oral mucosa showed the same trend as that of the patches to pig skin. For the group of PolyLA-Na/PolyLA-1.5-Y and PolyLA-Na/PolyLA-2-Y, the oral mucosa adhesion strength is enhanced with the increase of PolyLA content, because the adhesive carboxyl groups increases on the surface of adhesive patch, which can be proved by the XPS test (Fig. 3d-g in the original manuscript). But, for PolyLA-Na/PolyLA-3-Y group, the adhesion strength shows a trend of increase first and then decrease with the increment of the PolyLA content. The reason for this phenomenon is that with the increase of PolyLA content, although the surface carboxyl group of the patch is more abundant, the bulk strength of the patch is significantly reduced due to the amorphous structure (this can be confirmed by the tensile results of the patches with different compositions in Fig. 2g-l and Supplement Fig. 12 and 13 in the original manuscript), and the patch bulk is easily stretched when the adhesion strength is measured (**Fig. R9**), so it shows a reduced adhesion strength. In addition, based on the hydrophobic nature of PolyLA, the hydrophobicity of the patch increases with the increase of PolyLA content (this can be confirmed by the water contact angle test of the patch

with different compositions in the Fig. 3b and Supplement Fig. 14 in the original manuscript), so its ability to remove interface water of the adhered tissue decreases, which will also affect its adhesion strength. Therefore, the adhesion and content hypothesis we proposed in the manuscript can be confirmed by comprehensive analysis of the adhesion strength trend, XPS, tensile test results and water contact angle of the patch.

Fig. R9 Photographs showing the lap-shear process of the PolyLA-Na/PolyLA-3-1 patch adhered to pig oral mucosa tissue, and the PolyLA-Na/PolyLA-3-1 patch is highly stretched during test.

9.I do not see the methods used for the durable wet tissue adhesion test (supplementary fig 16) was this performed with continuous agitation for 1-24 hours? Why was pig skin and not pig oral mucosa used?

Response: We thank the reviewer's comment. According to the opinions of the reviewer, we replaced the pig skin with pig oral mucosal tissue to re-examine the durable adhesion of the patch to oral mucosal in a wet environment. We completely soaked the PolyLA-Na/PolyLA-2-1 patch adhered porcine oral mucosa in artificial saliva, and stirred at 37 °C at a speed of 120 r/min for 24 h. We found that the patch still firmly adhered to the oral mucosal tissue even in high wet and dynamic environment for a long time and could lift the weight of the mucosal tissue itself (**Fig. R5**). This indicates that the PolyLA-Na/PolyLA-2-1 patch has durable adhesion to oral mucosa. The test method has been supplemented in the revised manuscript and has been marked red.

Fig. R5 Durable adhesion ability of PolyLA-Na/PolyLA-2-1 patch to oral mucosa tissue.

10.Line 269–“polyLA-Na/PolyLA-based adhesive patches all exhibit negligible swelling for the first 4 h...” the data in fig 4a shows very rapid swelling for around half the formulations with large increases in swelling ratio so this statement does not represent the data–please amend.

Response: We thank the reviewer’s comment. We have modified the statement about the swelling of the patch and marked it red in the revised manuscript.

11.Release of PolyLA monomers following submersion in saliva–the data are provided as increases in absorbance in UV spectra and % release rate Over 50 min. Can the actual amount of LA-Na monomer released be calculated and presented as this data would be very informative?

Response: We thank the reviewer’s comment. In this work, we measured the monomer release of PolyLA-Na/PolyLA patches with different composition after soaking in artificial saliva for 48 h. The FT-IR results of the patch before and after soaking in artificial saliva showed that the characteristic absorption peak of $-C=O$ in carboxylates almost completely disappears; while the characteristic absorption peak of $-C=O$ at 1551 cm^{-1} on carboxyls is preserved (Fig. 4c in the original manuscript), indicating that the PolyLA-Na network in patch is almost completely dissociated, whereas the PolyLA network is retained, and the monomers released from patch are mainly LA-Na. However, from the results of monomer release efficiency, a small amount of TA monomers is also released from the patch, which may be due to the solubilization of Na^+ on LA. But we have no way to accurately calculate the ratio of LA to LA-Na in the released monomers; since saliva is a highly ionized solution, LA

will be quickly recombined into LA salts after release, which is difficult to distinguish from LA-Na.

12. ROS inhibition assessment. PolyLA-Na monomer alone should be used as a positive control. Can prior disruption of the PolyLA-NA monomer be performed before addition to DPPH to further show that ROS inhibition is specifically due to this chemical species?

Response: We are truly grateful to the reviewer's comments. In the original manuscript, we used ethanol and water as mixed solvents to determine the scavenging effect of PolyLA-Na/PolyLA patch on DPPH free radicals. We note that PolyLA-Na is insoluble in the mixture of ethanol and water. DPPH scavenging experiment cannot be done with PolyLA-Na. So in order to more truly simulate the application of PolyLA-Na/PolyLA patch in oral cavity, we re-determined the scavenging ability of PolyLA-Na/PolyLA patch on water soluble 2,2'-azino-bis(3-ethylbenzothiazoline-6-sulfonic acid) diammonium salt radical (ABTS) radicals in aqueous phase in the revised manuscript, and we selected the pure PolyLA-Na as the positive control and the PolyLA-Na/PolyLA-2-1 skeleton after soaking in water for 48 h as the negative control. As shown in **Fig. R10**, PolyLA-Na dissolves rapidly in aqueous solution and shows excellent antioxidant capacity, and its scavenging efficiency of ABTS free radicals can reach 90% in 5 h. On the other hand, PolyLA-Na/PolyLA-2-1 skeleton showed negligible antioxidant capacity, which could scavenge only 28% of ABTS free radicals at 12 h. Different from the two, PolyLA-Na/PolyLA patch can stably exist in aqueous solution and show continuous antioxidant capacity. Taking PolyLA-Na/PolyLA-2-1 as an example, its antioxidant efficiency can reach 75% at 5 h and 99% at 12 h. From the above results, it can be proved that the antioxidant ability of PolyLA-Na/PolyLA patch mainly benefits from the active molecule of LA-Na, which is released by the dissociation of PolyLA-Na network in the patch. And the PolyLA network in the patch can maintain the stability of the patch in the liquid environment and reduce the dissociation speed of the PolyLA-Na network. The relevant content has been supplemented in the revised

manuscript, and the revised part has been marked red.

Fig. R10 (a) ABTS radical clearance ratio of PolyLA-Na (Positive control), PolyLA-Na/PolyLA adhesive patches and PolyLA-Na/PolyLA-2-1 skeleton (Negative control) determined at different times; (b) Photographs of ABTS solutions in the absence of (control) and presence of ABTS/PolyLA-Na/PolyLA adhesive patches, ABTS/PolyLA-Na and ABTS/PolyLA-Na/PolyLA-2-1 skeleton after treatment for 12 h.

13. Fig 4f&g—what is the difference between images f and g (there appears to be no information in the methods concerning this experiment?). Why are the patches smaller in g than f? Also, this experiment contains no patch control. A positive control of PolyLA-Na monomer added in liquid form to an absorbent disc (used commonly in disc diffusion assays) would provide good evidence that the antibacterial effect is due to the released monomer and should be performed.

Response: We thank the reviewer's comment. We are very sorry that the explanation of Fig. 4f and g is missing in the original manuscript. We have added an explanation in the revised manuscript. Fig. 4f and g represent inhibition zone results for the PolyLA-Na/PolyLA adhesive patches against *E.coli* and *S.aureus* at 12 h and 24 h, respectively. The patch size in figure 4g and f is the same, but with the extension of time at the interface of the medium, the patch changes from yellow to transparent, so it is visually felt that the size of the patch has changed. We supplemented the control group without patch; as shown in **Fig. R11**, the paper soaked with PBS was placed in the medium and co-cultured with bacteria as a blank control. The results showed that there was no inhibition zone around the paper, indicating that the paper did not have

antibacterial activity, which also proved that our PolyLA-Na/PolyLA base patch had an obvious inhibitory effect on both Gram-positive bacteria and Gram-negative bacteria.

Fig. R11 Digital images displaying representative inhibition zone results for the PolyLA-Na/PolyLA adhesive patches against *E. coli* and *S. aureus* at 12 h (a) and 24 h (b), respectively.

In answering the 16th question raised by the reviewer, we tested the minimal inhibitory concentration (MIC) of PolyLA-Na/PolyLA-2-1 against Gram-positive bacteria and Gram-negative bacteria, and PolyLA-Na/PolyLA-2-1 patch skeleton was selected as the control. As shown in **Fig. R12**, the antibacterial efficiency increased with the increase of patch concentration after co-culturing with bacteria for 24 h. Moreover, the MIC of PolyLA-Na/PolyLA-2-1 patch against both Gram-positive bacteria and Gram-negative bacteria was 12 mg/mL, in which about 6 mg/mL of LA-based small molecules were released. However, unlike the PolyLA-Na/PolyLA-2-1 patch, the PolyLA-Na/PolyLA-2-1 patch skeleton shows negligible antibacterial properties, which indicates that the antibacterial activity of the

patch mainly depends on the dissociation of the PolyLA-Na network and the release of LA-Na monomers in the patch.

Fig. R12 (a) Photographs of bacterial cultures in different mass concentration PolyLA-Na/PolyLA-2-1 patch and PolyLA-Na/PolyLA-2-1 patch skeleton groups; (b) Antibacterial efficiency of different mass concentration of PolyLA-Na/PolyLA-2-1 patch and PolyLA-Na/PolyLA-2-1 patch skeleton against *E. coli* after co-culturing for 24 h; (c) Antibacterial efficiency of different mass concentration of PolyLA-Na/PolyLA-2-1 patch and PolyLA-Na/PolyLA-2-1 patch skeleton against *S. aureus* after co-culturing for 24 h.

14. Gram-positive and Gram-negative bacteria have completely different cell walls, although it seems from fig h & j that inhibition of growth is instantaneous—can the authors explain why this is so—do the cells undergo instantaneous cell lysis? What is the mechanism of action? In addition, fig.4 h/j and I/k do not match—in h/j growth is completely inhibited (0-5 min) while in i/k there is only 60% inhibition at 2 hours—why is this?

Response: We thank the reviewer’s comment. The abscissa units of Figure 4h and j in the original manuscript are hours rather than minutes. In this work, the PolyLA-Na/PolyLA-based patch does not lead to immediate inhibition of bacterial proliferation after contact with bacteria, but inhibits bacterial proliferation with the

gradual release of LA-Na monomer in the culture medium. Therefore, the bacteria were not completely inhibited in the short time of contact with the PolyLA-Na/PolyLA based patch, but proliferated slowly, and the proliferation rate was much lower than that of the control group without adding patch.

The antibacterial activity of the patch mainly depends on the release of LA-Na-based small molecules. It has been proved in the previous reports that LA-based active small molecules have inhibitory effect on both Gram-positive and Gram-negative bacteria.^[41-44] Although there is a great difference in the composition of cell wall between Gram-positive bacteria and Gram-negative bacteria, the antibacterial mechanism of LA-based small molecules against Gram-positive bacteria and Gram-negative bacteria is the same, which is based on the strong depolarization of LA-based small molecules to bacterial cell membrane to change the permeability of bacterial cell membrane and make the internal substances of bacteria leak out, resulting in their death. In addition, LA-based small molecules can also affect the metabolic disorder of bacteria, and then inhibit the bacteria proliferation. We have added the relevant content in the revised manuscript, and marked it red.

15.Line 329–Gram-positive.

Response: We thank the reviewer's comment. We have corrected it in the revised manuscript.

16.The minimum inhibitory concentration for the patch eluent against E. coli and S. aureus should be determined as this provides a better measure of antimicrobial activity?

Response: We thank the reviewer's comment. As shown in question 14, we tested the minimum inhibitory concentration (MIC) of the PolyLA-Na/PolyLA-2-1 patch and added it to the revised manuscript.

17. The authors should state why chitosan was used as a control film (and also state the company where these patches were sourced in the materials and methods). It is

difficult to determine the presence of the PolyLA-Na/PolyLA2-1 patch in the rat oral cavity at 6-24 h in fig 5d although this might be due to image quality. This needs to be clarified.

Response: We thank the reviewer's comment. In this work, we chose chitosan patch as the control material because chitosan patch is a commonly used commercial oral ulcer repair patch with certain antibacterial ability, and the chitosan patch we use in this work was purchased from Chuangbang Medical and Health Technology (Yunnan, China) Co., Ltd. We have added the relevant information in the revised manuscript. We are very sorry for your confusing caused by the poor quality of the image; we provide a clear image and enlarge the adhesion state of the patch in the mouth at different times. As shown in **Fig. R13**, the PolyLA-Na/PolyLA-2-1 patch can firmly adhere to the oral mucosa of rats, and the patch volume decreases with the extension of time in the oral cavity, which may be due to the in-situ degradation of the patch caused by reductase in the oral cavity. After 12 h in the oral cavity, the patch still adhered closely to the oral mucosa, and the patch completely degraded in the oral cavity at 24 h.

Fig. R13 Images of the durable adhesion of PolyLA-Na/PolyLA-2-1 patch in oral cavity.

18. In vitro toxicity on fibroblast monolayers—was the same eluent concentration (1

mg/ml) used in these experiments as were conducted for the antimicrobial experiments? Did the authors test toxicity toward oral keratinocytes? In addition, biocompatibility should be investigated using cells in direct contact with the material rather than only following exposure to the supernatant. For this tissue-engineered oral mucosal models should be used. These are commercially available and are much more representative of the oral mucosa than cells cultured as monolayers and toxicity experiments performed to OECD standards. The biopsy histological data for rat mucosal tissue would back up this data to confirm low toxicity.

Response: We thank the reviewer's comment. In this work, the antibacterial concentration of the patch is not consistent with that of the cytotoxicity test. As mentioned above, we investigated the bacteriostatic effect of different concentrations of patch and confirmed that the patch had good inhibitory effect on both Gram-positive bacteria and Gram-negative bacteria. Moreover, the in vivo experiment also proved that the antibacterial activity of PolyLA-Na/PolyLA-based patch was better than that of commercial chitosan oral repair patch. Therefore, even if the concentration of PolyLA-Na/PolyLA-based patch in antibacterial experiment is higher than that in cell experiment, its effective antibacterial effect cannot be denied.

In this work, we examined the cytotoxicity of the patch to gingival fibroblasts rather than oral keratinocytes. This is because after the formation of oral ulcer, the stratum corneum of the ulcer is destroyed, and the patch comes into contact with fibroblasts and endothelial cells, so it is reasonable to investigate the compatibility of the patch with gingival fibroblasts.^[26,28,35] In order to further prove the cytocompatibility of the patch, we investigated the effect of the patch on the cell viability of human umbilical vein endothelial cells (HUVECs). The results showed that the patch also had very good cytocompatibility to HUVECs, and it can obviously promote the proliferation of endothelial cells after co-culture for 3 days (**Fig. R14**). It is proved once again that the patch has good cell compatibility and the ability to promote ulcer tissue repair. In addition, the oral cavity is a wet environment rich in saliva. When the PolyLA/PolyLA-Na patch is used in the oral cavity, saliva dissociates the PolyLA-Na network in the patch to release LA-Na active molecules,

thus accelerating ulcer healing. Therefore, in the cell experiment, it is reasonable for us to use the extract to investigate the cytocompatibility of the patch. Moreover, the concentration of LA-based monomer in the extract is much higher than that of LA-based monomer in blood of rats (Table S1 in the original manuscript), which indicated that the clinical dosage of PolyLA/PolyLA-Na adhesive patch is cytocompatible. The biosafety of PolyLA/PolyLA-Na adhesive patch can also be proved by histological staining data of rat oral mucosa. The H&E staining results show that compared with the normal oral mucosa tissue, there is no significant inflammation, necrosis, or metaplasia in the buccal mucosa tissue treated with the PolyLA-Na/PolyLA-2-1 adhesive patch for 12 h (Fig. 5g in the original manuscript), indicating that PolyLA-Na/PolyLA-2-1 adhesive patch has low toxicity and stimulates no acute reaction for oral mucosa.

Fig. R14 Cell viability of HUVECs cells after co-cultured with PolyLA-Na/PolyLA patch for 1, 2, 3 days (n=4).

19. Line 396—“PolyLA-Na/PolyLA group is smoother” ...this is a subjective term that is not measured parameter, suggest changing this term.

Response: We thank the reviewer’s comment. We have modified the description of this part in the revised manuscript.

20. How was the wound size and rate of closure calculated as this would depend on the angle the image was taken? Wounds treated with PolyLA-patch and imaged at 2 d displayed variable would healing rate but this variability had dramatically declined at

4 d—can the authors explain this loss of experimental variability? Did the authors perform a power calculation to determine population size for rodent numbers in experiments (this is n=3 in the manuscript).

Response: We thank the reviewer's comment. It is a general method to calculate the wound healing efficiency by taking photos for wound and recording the wound area.^[26,27,35,45] Although there does exist the influence of photo angle in this method, we try to keep the wound at the same angle when taking pictures to reduce the error. In addition, by comparing the collagen deposition, the degree of epithelialization and the distribution of inflammatory factors in different groups, our PolyLA-Na/PolyLA patch is better than commercial chitosan patch in accelerating the repair of oral ulcer.

The wound healing cascade typically encompasses three programmed but overlapping phases including inflammation, cell proliferation, and tissue remodeling^[46-48]. The early stage of wound healing is the period of inflammatory reaction of the wound, which is dominated by the infiltration of inflammatory cells, so there is little difference in the efficiency of wound healing among different groups. After the inflammatory stage, the excellent biological activity of PolyLA-Na/PolyLA patch can effectively regulate the biological behaviors of host cells, such as cell migration, infiltration, and differentiation, and promoting the secretion of growth factors. Moreover, based on the bioactivity of PolyLA-Na/PolyLA patch, it can also effectively regulate the microenvironment of the wound, eliminate bacterial infection, promote the formation of blood vessels and epithelium, and then accelerate tissue remodeling and regeneration. In short, PolyLA-Na/PolyLA patch plays a key role in both cell proliferation and tissue remodeling in the process of wound healing. Therefore, on the fourth day, the wound healing efficiency of the PolyLA-Na/PolyLA patch group was significantly higher than that of the other two groups.

In our work, we only preliminarily explored the effect of PolyLA-Na/PolyLA patch in accelerating the repair of oral mucosal injury through animal experiments, so the rodent numbers in the vivo experiments were not determined by powder calculation, but by referring to protocol published in literature.^[27,35] In the follow-up clinical trials, we will determine the number of experimental animals in vivo through

powder calculation.

21. Fig 6 d/e/f the PolyLA-patch shows a remarkable effect on acid induced rodent ulcers which is very encouraging. The authors attempt to link this wound healing property to inhibition of leukocyte recruitment in patch-treated rats but this analysis is not robust. The immunostaining particularly for iNOS/MPO looks very non-specific and the images are of poor quality. There is no information on how this analysis was done—how many fields of view and sections analysed per rodent? Was the analysis done blind? The statistics state that ANOVA was used but how is the data normally distributed when n=3?

Response: We thank the reviewer's comment. We are very sorry for the trouble caused to the reviewer because of the poor quality of the pictures. Here we provide high-resolution immunofluorescence results (**Fig. R15**), from which we can see that the expression of inflammation-related M1 macrophages and neutrophils in PolyLA-Na/PolyLA group is significantly lower than that in chitosan group and control group. Moreover, the results of H&E staining also showed that there was obvious inflammatory cell infiltration in the chitosan group and the control group (Fig. 6d in the original manuscript). These results can all prove that PolyLA-Na/PolyLA patch has an excellent anti-inflammatory effect.

Fig. R15 Immunofluorescence staining and quantification expression levels of iNOS and MPO antibody in regenerated oral mucosa at 4 days.

In this work, three fields of view per section were analyzed for iNOS and MPO staining, and one field of view per section was analyzed for CK5, CK13, CD31 and Sirius red staining, and all results were analyzed by a pathologist who were blinded to experimental design. And the student's t-test was used to examine the differences between the two groups of data rather than the variance test.

22. There is some good microbiology work conducted that shows differences between PolyLA-patch and chitosan-patch treatments. However, why would the microbiota in the control group who have had treatment with acid (a treatment designed to kill cells, including bacteria, thus creating an ulcer) be the same as in the normal (untreated) group. This does not seem intuitive? Can the authors comments on this.

Response: This is a good question. We speculated that the reason for this result may be that although the use of acid etching method to make oral ulcer model can cause oral mucosal damage, it cannot significantly destroy the oral microflora, which is different from microbial-infected oral ulcers. When the antibacterial or anti-inflammation material is used in the oral cavity, it is very easy to cause the risk of oral microbiota imbalance and double infection.^[49-52] From the results of the changes of oral microbiota in Fig. 7 in the original manuscript, we can see that after the use of chitosan patch, the oral microbiota changed significantly compared with the normal group, and even the number of some pathogenic bacteria increased obviously, indicating that chitosan caused the imbalance of oral microbiota. However, after using our PolyLA-Na/PolyLA patch, the microbiota in the oral cavity was similar to that in the normal group, indicating that PolyLA-Na/PolyLA would not cause changes in the microbiota in the oral cavity. Therefore, this result can still evaluate the effect of materials on the microecology of oral microbiota, and can prove that compared with commercial chitosan patch, PolyLA-Na/PolyLA patch can not only eliminate harmful bacteria but also avoid the destruction of oral microflora.

23. In figure 8b, what is the yellow covering the wound in the control on day 3? This

looks like a wound covering but there should not be any in this control pig?

Response: We thank the reviewer's comment. The yellow covering on the wound is not the wound covering, but the exudate and scab tissue of the wound during the repair of oral mucosa. It can also be seen from the pictures (**Fig. R16**) that compared with the PolyLA-Na/PolyLA-based patch group, the control group and chitosan group secrete more darker exudates, which also proves that our patch has better anti-inflammatory effect.

Fig. R16 Photographs of buccal mucosa defect in mini-pig treated by PolyLA-based patch and chitosan film at the 0, 3, 5, and 8 days.

24. How did the authors take into account the angle of the wound images in order to calculate the healing ration. For example, images at day 5 for control, PolyLA patch and Chitosan are taken at different angles which appear to show that the control has a larger wound size whereas the treated images are taken at a shallower angle, reducing the wound area?

Response: We thank the reviewer's comment. Because of the pig's long mouth, the shooting angle when taking pictures of the wound may indeed have an impact on the evaluation of wound healing efficiency, but we have tried to maintain the same shooting angle to reduce this error, and we have made statistics through multiple wounds to reduce the impact of shooting angle. In addition, on the 8th day, the healing efficiency of PolyLA-Na/PolyLA patch group was significantly better than that of chitosan patch group and control group, and histological analysis also proved

that PolyLA-Na/PolyLA patch could better accelerate the healing of oral ulcer.

25. The H&E and Masson-stained images look encouraging and strongly point towards quicker re-epithelialisation. However, the analysis of other markers such as Sirius red, CD31 and iNOS, IL-6, TNF is not robust and not well-defined in the methods section (e.g., much non-specific staining and the images are of poor quality. There is no information on how this analysis was done—how many fields of view and sections analysed per rodent? Was the analysis done blind? The statistics state that ANOVA was used but how is the data normally distributed when $n=3$?). Moreover, the potential effect on cytokine expression and angiogenesis is highlighted as a novel finding but these are based on observational findings. Further experiments are required here (e.g., tubule formation assays, in vitro endothelial cell, migration and proliferation assays, CAM assay, TNF/IL6/collagen gene expression assay on various cell types such as in vitro cultured macrophages, fibroblasts with and without Poly-LA eluant) are required here to confirm these initial preliminary observations.

Response: We thank the reviewer's comment. We are very sorry for the trouble caused to the reviewers because of the poor quality of the pictures. Here we provide high-resolution Sirius red staining and immunofluorescence results of oral mucosa healing tissue of mini-pigs (**Fig. R17-R19**), in which Sirius red is Specific staining for the collagen, CD31 is staining for the formation of new blood vessels, immunofluorescence staining of iNOS and MPO to evaluate the percentage of inflammatory cells, while IL-6 and TNF is staining for inflammatory factors. The results of Sirius red staining show that the well-organized collagen fibers in PolyLA-Na/PolyLA patch group were seen, while an obvious defect of collagen fibers was noticed in chitosan and control group. And from the results of immunofluorescence staining, we can see that PolyLA-Na/PolyLA patch group has more neovascularization and lower distribution of inflammatory cells and inflammatory factors, indicating that compared with chitosan patch group and control group, PolyLA-Na/PolyLA has the function of promoting blood vessels and anti-inflammation.

In this work, three fields of view per section were analyzed for iNOS and MPO staining, and one field of view per section was analyzed for CK5, CK13, CD31 and Sirius red staining, and all results were analyzed by a pathologist who were blinded to experimental design. And the student's t-test was used to examine the differences between the two groups of data rather than the variance test.

Fig. R17 Sirius red staining of type I collagen (red) and type III collagen (green) in the regenerated mini-ping oral mucosa at 8 days.

Fig. R18 Immunofluorescence staining of CK5, CD31 and iNOS antibody in regenerated oral mucosa at 8 days.

Fig. R19 Immunohistochemical staining of IL-6 and TNF- α antibody in regenerated oral mucosa at 8 days.

In order to further verify that the active small molecules released by PolyLA-Na/PolyLA patch can promote cytokine expression and angiogenesis at the cellular level, we investigated the ability of PolyLA-Na/PolyLA patch to promote HUVECs cell migration and tube formation. As shown in **Fig. R20**, the HUVECs cell migration of Patch 1 and Patch 2 group was faster than that of the control group. Some scratch area was still observed in control group, while a complete closure was seen in the Patch 1 and Patch 2 group at 24 h. Quantitative analysis displayed that migration rate of Patch 1 and Patch 2 group was 69% and 75% at 12 h and 98% and 99% at 24 h, respectively, which were much higher than those in control group (57% at 12 h, 88% at 24 h). The results show that PolyLA-Na/PolyLA patch can effectively promote endothelial cell migration, which is very beneficial to promote wound healing^[26].

Fig. R20 Representative images of HUVECs cell migration at different times and the migration rate of HUVECs cells co-cultured with PolyLA-Na/PolyLA patch for different time ($n = 3$) (Patch-1 is $62.5 \mu\text{g/mL}$, Patch-2 is $31.25 \mu\text{g/mL}$).

HUVECs tube formation assay was also performed to investigate the pro-angiogenesis potential of the PolyLA-Na/PolyLA patch. As shown in **Fig. R21**, compared with control group, more integrated and diverse well-branched tubes formed in in Patch 1 and Patch 2 group. The number of capillary-like structures in patch 1 and patch 2 was much higher than that in the control group, indicating that the PolyLA-based patch has great potential pro-angiogenesis.

Fig. R21 Representative images of capillary-like structures of HUVECs and

quantification of formed tube number of HUVECs cells co-cultured with PolyLA-Na/PolyLA patch (n = 4).

The in vitro anti-inflammatory effect of the PolyLA-Na/PolyLA patch was evaluated by coculturing with LPS stimulated Mouse Monocyte-macrophage Leukemia Cells (RAW264.7). As shown in **Fig. R22**, pro-inflammatory cytokines TNF-a and IL-6 increased significantly after LPS stimulation, but after PolyLA-Na/PolyLA treatment, the production levels of the two pro-inflammatory factors decreased obviously, and the decreasing effect was positively correlated with the patch concentration. These results indicated that the PolyLA-Na/PolyLA adhesive patch exhibited strong anti-inflammatory effect on LPS stimulated RAW264.7 cells.

Fig. R22 Effect of PolyLA-Na/PolyLA patch on TNF-a and IL-6 production in LPS stimulated RAW264.7 cells.

To sum up, our PolyLA-Na/PolyLA patch showed obvious anti-inflammatory and vascular-promoting effects both in vitro and in vivo, which laid a foundation for accelerating the repair of oral mucosal injury. All supplement contents have been added to the revised manuscript and have been marked red.

26. The discussion is just a reiteration of the results, it does not place the study in context with other contemporary studies.

Response: We thank the reviewer's comment. We have modified the part of discussion in the revised manuscript and marked it red.

References:

- [1] Liu, X., Zhang, Q. & Gao, G. H. Bioinspired adhesive hydrogels tackified by nucleobases. *Adv. Funct. Mater.* **27**, 1703132 (2017).
- [2] Pei, X. J. et al. Stretchable, self-healing and tissue-adhesive zwitterionic hydrogels as strain sensors for wireless monitoring of organ motions. *Mater. Horiz.* **7**, 1872-1882 (2020).
- [3] Li, S. Z. et al. Self-healing hyaluronic acid nanocomposite hydrogels with platelet-rich plasma impregnated for skin regeneration. *ACS Nano* **16**, 11346-11359 (2022).
- [4] Drotlef, D. M. et al. Insights into the adhesive mechanisms of tree frogs using artificial mimics. *Adv. Funct. Mater.* **23**, 1137-1146(2023).
- [5] Kim, D. W. et al. Electrostatic-mechanical synergistic in situ multiscale tissue adhesion for sustainable residue-free bioelectronics interfaces. *Adv. Mater.* **34**, 2105338(2021).
- [6] Jeon, I. et al. Extremely stretchable and fast self-healing hydrogels. *Adv. Mater.* **28**, 4678(2016).
- [7] Cui, C. Y. et al. Water-triggered hyperbranched polymer universal adhesives: from strong underwater adhesion to rapid sealing hemostasis. *Adv. Mater.* **31**, 1905761(2019).
- [8] Yang, J. W., Bai, R. B. & Suo Z. G. Topological adhesion of wet materials. *Adv. Mater.* **30**, 1800671(2018).
- [9] Gao, Y. et al. A universal strategy for tough adhesion of wet soft material. *Adv. Funct. Mater.* **30**, 2003207(2020).
- [10] Yuk, H. et al. Dry double-sided tape for adhesion of wet tissues and devices. *Nature* **575**, 169-178(2019).
- [11] Wu, S. J. et al. A multifunctional origami patch for minimally invasive tissue sealing. *Adv. Mater.* **33**, 2007667(2021).
- [12] Zhao, Q. et al. Underwater contact adhesion and microarchitecture in polyelectrolyte complexes actuated by solvent exchange. *Nat. Mater.* **15**, 407-412(2016).
- [13] Liu, Y. et al. A moldable nanocomposite hydrogel composed of a mussel-inspired polymer and a nanosilicate as a fit-to-shape tissue sealant. *Angew. Chem. Int. Ed.* **56**, 4224-4228(2017).
- [14] Han, L. et al. Mussel-inspired adhesive and tough hydrogel based on nanoclay confined dopamine polymerization. *ACS Nano* **11**, 2561-2574(2017).
- [15] Liang, Y. Q. et al. Dual-dynamic-bond cross-linked antibacterial adhesive hydrogel sealants

with on-demand removability for post-wound closure and infected wound healing. *ACS Nano* **15**, 7078-7093(2021).

[16] Chen, C. et al. Tannic acid-thioctic acid hydrogel: A novel injectable supramolecular adhesive gel for wound healing. *Green Chem.* **23**, 1794-1804 (2021).

[17] Lv, S. Y. et al. Review of lipoic acid: from a clinical therapeutic agent to various emerging biomaterials. *Int. J. Pharmaceut.* **627**, 122201 (2022).

[18] Zhou, W. Y. et al. Lipoic acid modified antimicrobial peptide with enhanced antimicrobial properties. *Bioorgan. Med. Chem.* **28**, 115682 (2020).

[19] Li, W. T. et al. α -Lipoic acid stabilized DTX/IR780 micelles for photoacoustic/fluorescence imaging guided photothermal therapy/chemotherapy of breast cancer. *Biomater. Sci.* **6**, 1201-1216(2018).

[20] Li, M. et al. A reduction-responsive drug delivery with improved stability: disulfide crosslinked micelles of small amphiphilic molecules. *RSC Adv.* **11**, 12757-12770(2021).

[21] Wu, Z. Z. et al. Using copper sulfide nanoparticles as cross-linkers of tumor microenvironment responsive polymer micelles for cancer synergistic photo-chemotherapy. *Nanoscale* **13**, 3723-3736(2021).

[22] Zhang, Q. et al. Exploring a naturally tailored small molecule for stretchable, self-healing, and adhesive supramolecular polymers. *Sci. Adv.* **4**, eaat8192(2018).

[23] Wang, Y. J., Sun, S. T. and Wu, P. Y. Adaptive ionogel paint from room-temperature autonomous polymerization of α -thioctic acid for stretchable and healable electronics. *Adv. Funct. Mater.* **31**, 2101494(2021).

[24] Khan, A. et al. Highly stretchable supramolecular conductive self-healable gels for injectable adhesive and flexible sensor applications. *J. Mater. Chem. A* **8**, 19954-19964(2020).

[25] Zhang, Q. et al. Assembling a natural small molecule into a supramolecular network with high structural order and dynamic functions. *J. Am. Chem. Soc.* **141**, 12804-12814(2019).

[26] Zhang, W. J. et al. Promoting Oral mucosal wound healing with a hydrogel adhesive based on a phototriggered s-nitrosylation coupling reaction. *Adv. Mater.* **33**, 2105667(2021).

[27] Hu, S. S. et al. A mussel-inspired film for adhesion to wet buccal tissue and efficient buccal drug delivery. *Nat. Commun.* **12**, 1689(2021).

[28] Xing, J. Q. et al. Barnacle-Inspired robust and aesthetic Janus patch with instinctive wet

- adhesive for oral ulcer treatment. *Chem. Eng. J.* **444**, 136580(2022).
- [29] Liu, X. Q. et al. Designing a mucoadhesive chemoPatch to ablate oral dysplasia for cancer prevention. *Small* **18**, 2201561(2022).
- [30] Paris, A. L. et al. Sublingual protein delivery by a mucoadhesive patch made of natural polymers. *Acta Biomater.* **128**, 222-235(2021).
- [31] Colley, H. E. et al. Pre-clinical evaluation of novel mucoadhesive bilayer patches for local delivery of clobetasol-17-propionate to the oral mucosa. *Biomaterials* **178**, 134-146(2018).
- [32] Edmans, J. G. et al. Electrospun patch delivery of anti-TNF α F (ab) for the treatment of inflammatory oral mucosal disease. *J. Control. Release* **350**, 146-157(2022).
- [33] Teno, J. et al. Development of multilayer ciprofloxacin hydrochloride electrospun patches for buccal drug delivery. *J. Funct. Biomater.* **13**, 170(2022).
- [34] Zhang, H. et al. Fabrication of astaxanthin-loaded electrospun nanofiber-based mucoadhesive patches with water-insoluble backing for the treatment of oral premalignant lesions. *Mater. Design* **223**, 111131(2022).
- [35] An, H. et al. Janus mucosal dressing with a tough and adhesive hydrogel based on synergistic effects of gelatin, polydopamine, and nano-clay. *Acta Biomater.* **149**, 126-138(2022).
- [36] Zhu, J. J. et al. Low-swelling adhesive hydrogel with rapid hemostasis and potent anti-inflammatory capability for full-thickness oral mucosal defect repair. *ACS Appl. Mater. Interfaces* **14**, 53575-53592(2022).
- [37] Wang, X. et al. Hydrogel-based patient-friendly photodynamic therapy of oral potentially malignant disorders. *Biomaterials* **281**, 121377(2022).
- [38] Shi, C. Y. et al. Highly ordered supramolecular assembled networks tailored by bioinspired H-bonding confinement for recyclable ion-transport materials. *CCS Chem.* DOI: 10.31635/ceschem.022.202202158 (2022).
- [39] Taka, J. I., Ogino, S. & Kashino, S. Ammonium hydrogen tartrate. *Acta Cryst.* **54**, 384-386 (1998).
- [40] D'Ascenzo, L. & Auffinger, P. A comprehensive classification and nomenclature of carboxyl-carboxyl(ate) supramolecular motifs and related catemers: implications for biomolecular systems. *Acta Cryst.* **71**, 164-175 (2015).
- [41] Shi, C. et al. Antimicrobial effect of lipoic acid against *Cronobacter sakazakii*. *Food Control*

59, 352-358 (2016).

[42] Yang, S. Q. et al. Metabolomics analysis and membrane damage measurement reveal the antibacterial mechanism of lipoic acid against *Yersinia enterocolitica*. *Food Funct.* **13**, 11476-11488 (2022).

[43] Chen, Y. J., Yuan, T. & Liu, Z. Z. Supramolecular medical antibacterial tissue adhesive prepared based on natural small molecules. *Biomater. Sci.* **8**, 6235-6245 (2020).

[44] Zhou, W. Y. et al. Lipoic acid modified antimicrobial peptide with enhanced antimicrobial properties. *Bioorgan. Med. Chem.* **28**, 115682 (2020).

[45] Choi, S. et al. Mucoadhesive phenolic pectin hydrogels for saliva substitute and oral patch. *Adv. Funct. Mater.* 2303043(2023).

[46] Liu, W. S. et al. ECM-mimetic immunomodulatory hydrogel for methicillin-resistant *Staphylococcus aureus*-infected chronic skin wound healing. *Sci. Adv.* **8**, eabn7006 (2022).

[47] Kharaziha, M., Baidya, A. & Annabi, N. Rational design of immunomodulatory hydrogels for chronic wound healing. *Adv. Mater.* **33**, 2100176(2021).

[48] Blacklow, S. O. et al. Bioinspired mechanically active adhesive dressings to accelerate wound closure. *Sci. Adv.* **5**, eaaw3963(2019).

[49] Kopra, E. et al. Systemic antibiotics influence periodontal parameters and oral microbiota, but not serological markers. *Front. Cell. Infect. Microbiol.* **11**, 774665(2021).

[50] Pauter-Iwicka, K. et al. Characterization of the salivary microbiome before and after antibiotic therapy via separation technique. *Appl. Microbiol. Biot.* **107**, 2515-2531(2023).

[51] Liu, H. et al. Anti-infection mechanism of a novel dental implant made of titanium-copper (TiCu) alloy and its mechanism associated with oral microbiology. *Bioact. Mater.* **8**, 381-395(2022).

[52] Cheng, X. et al. Effects of none-steroidal anti-inflammatory and antibiotic drugs on the oral immune system and oral microbial composition in rats. *Biochem. Bioph. Res. Co.* **507**, 420-425(2018).

Reviewers' Comments:

Reviewer #1:

Remarks to the Author:

The authors have made significant efforts to address my comments. I would be delighted to recommend the publication of this work as it stands

Reviewer #2:

Remarks to the Author:

The authors developed biomaterials loading drugs or cytokines to therapy oral uicer. It is an interesting thing. After revision, the manuscript has attained the level of publication. So, I recommend accepting this paper for publication in NC.

Reviewer #3:

Remarks to the Author:

The authors have made changes to the manuscript in line with many of the reviewers comments. I feel that these have improved the clarity of the manuscript and have added further important information to the study, which, in my opinion, is now in a position to be recommended for publication.